# Cellular mechanical properties in response to environmental viscosity imaged by Brillouin Microscopy
Chenchen Handler[1,2], Giulia Zanini [3,5], Ian M. Smith[3], Kimberly M. Stroka[3], Giuliano Scarcelli [3] & Claudia Testi [3,4] ✉

An increasing body of research in biomechanics has revealed that the stiffness of the surrounding environment influences cells fate and function. In this context, a recent study showed that cells exposed to highly viscous fluids migrated and spread faster: the viscosity of the surrounding environment thus emerges as a novel potential regulator of key cell functions. To date, however, cellular mechanical responses to this biophysical trigger are widely unconsidered. In this study, we evaluate the mechanical properties of non-cancerous (MCF10A) and highly metastatic cancer (MDA-MB-231) cells grown in fluids of different viscosities by using our custom-built Brillouin Microscope. To achieve this result, we prove that the linewidth of a Brillouin spectrum can serve as a reliable viscosity indicator through an innovative deconvolution method that makes use of a Brillouin Microscope and a Stimulated Brillouin Microscope. Our findings suggest that cancer cells may adapt their internal mechanical properties in response to external media viscosity, thus improving their adaptability to the environment in an active interaction with their surroundings.

Cells experience a wide range of mechanical forces in vivo, such as hydrostatic pressures, interstitial compression, tensile and hydrodynamic forces[1–3]. Using a myriad of receptors coupled to intracellular organelles, cells detect and react to these external mechanical stimuli in a process known as mechanotransduction. This process converts forces into biochemical signals through complex cellular pathways, which in turn regulate tissue morphogenesis and differentiation[2–4]. Not restricted to normal homeostasis, alterations in cell and tissue mechanical properties can be indicative of the presence of pathologies. For example, metastatic cancer cells exhibit, albeit context-dependent, changes in stiffness[1,5–8], mutations of proteins governing chromatin relaxation correlate with increased nuclear stiffness[9–11], and extracellular matrix stiffness in tissues is strongly modified in fibrosis[12–14]. We and others have focused on the concomitant changes in stiffness associated with pathologies such as cancers[15–17]; however, the extent to which cells perceive the viscosity of their surrounding environment remains largely underexplored. A major limit to these studies is due to the available technology to assess the viscosity of biological samples under in vivo conditions: standard methods (e.g., AFM, micropipette aspiration, and micropillar deformation) are, in fact, contact-based techniques that can introduce invasiveness or spatial inaccuracies[5]. Moreover, culture conditions in vitro are employed using media with a viscosity similar to water (0.77 centipoise (cP)[18]), whereas physiological fluids such as blood, mucus or saliva typically have much higher viscosities (~2–4 cP), and under pathological conditions they can even exceed 8 cP[4,19]. Thus, one effort is to reconcile cellular response to this important environmental perturbation[20].

Recently[20–22], supraphysiological extracellular viscosities (>40 cP) have been shown to increase velocity and migration of both normal and cancer cells. Increasing external fluid viscosity (from 50 to 2000 cP) correlated with higher cellular velocity within a few hours of incubation. This result is surprising, as higher viscosity should impede the motion of particles in fluids. These observations highlight the active nature of cells and their dynamic response to environmental changes. Higher fluid viscosity was also found to enhance cellular spread area, focal adhesion, and traction forces[22]. Moreover, increases in extracellular viscosity at the primary tumor site conferred metastatic cells with advantages with respect to normal cells: cancer cells were found to retain memory of the different mechanical properties of extracellular fluids to which they were exposed during the metastatic cascade[15,21,23].

[1]Department of Mechanical Engineering, A. James Clark School of Engineering, University of Maryland, College Park, MD, USA. [2]Laboratory of Cell Biology, Center for Cancer Research, National Cancer Institute, National Institutes of Health, Bethesda, MD, USA. [3]Fischell Department of Bioengineering, A. James Clark School of Engineering, University of Maryland, College Park, MD, USA. [4]Center for Life Nano- and Neuro-Science, Istituto Italiano di Tecnologia, Viale Regina Elena 291, Roma, Italy. [5]Present address: CrestOptics S.p.A., Via Di Torre Rossa 66, Roma, Italy. ✉e-mail: claudia.testi@iit.it

Extracellular fluid viscosity, thus, could act as a potential regulator of relevant cell functions. In this context, we aimed to investigate if and how extracellular fluid viscosity regulates cellular mechanical properties. In particular, our study aims to investigate whether breast cells of different cancer stages (namely, MCF10A as non-cancerous breast cells and MDA-MB-231 as highly metastatic breast cancer cells[24]) sense varying media viscosities. To achieve cellular mechanical characterization in living cells, we employed Brillouin microscopy, a novel label-free technique sensitive to the longitudinal modulus and longitudinal viscosity at high frequencies. Brillouin microscopy can resolve mechanical properties of living cells with sub-micrometric resolution in a non-contact manner[12,25,26]. The role of viscosity in Brillouin microscopy remains unclear and has not been explored much in the current literature, where mainly Brillouin shift maps are shown to assess sample stiffness: the Brillouin spectra are indeed subjected to frequency broadening coming from the spectrometer response function, thus necessitating a spectral deconvolution to extract the intrinsic Brillouin linewidth of the sample. For this reason, it has been challenging to accurately measure the Brillouin linewidth using frequency-domain approaches[27–32]; nonetheless, when appropriate deconvolution is performed, the Brillouin spectra have demonstrated strong sensitivity to the material's longitudinal viscosity[33].

In this work, we: (i) propose a methodology to extract the intrinsic Brillouin linewidth of inorganic and biological samples, related to their local viscosity, from the full width at half maximum (FWHM) of their spectra acquired with our spontaneous Brillouin microscope; (ii) characterize MCF10A and MDA-MB-231 Brillouin shifts and deconvolved FWHMs to estimate both cellular stiffness and viscosity of cells grown in normal (0.77 cP) and highly viscous (8 cP) culture media.

In such a manner, we gained a more comprehensive understanding of how extracellular viscosity influences cellular mechanical properties and how intracellular viscosity varies with metastatic progression.

## Results
### Deconvolution of Brillouin spectra to extract sample viscosity with calibration materials

Mechanical moduli are complex and frequency-dependent functions, i.e., they have the form $A(v)^* = A(v)' + iA(v)''$. Here, the real and the imaginary parts characterize the responses of the material upon an external perturbation: the former characterizes the elastic behavior, related to the sample modulus, while the latter represents the dissipative response, related to the sample viscosity. From the Brillouin spectrum (Fig. 1A, left panel), it is possible to quantify the complex Longitudinal Modulus $M$, composed of a real part ($M'$) and an imaginary part ($M''$). This modulus represents the ratio of uniaxial stress to strain whereby the material experiences a change in volume, different from the more commonly used Young's modulus, where the material volume is kept constant. Moreover, because Brillouin probes material properties at high frequencies (GHz regime), the measured modulus is reported in the GPa range; other mechanical testing methods (i.e., atomic force microscopy, AFM) use much lower probing frequencies and the measured modulus is in the Pa–kPa range[26]. While the longitudinal modulus $M$ and Young's modulus $E$ are fundamentally different, they both follow the same direction of deformation, and empirical relationships between $M$ and $E$ have been established for specific experimental conditions[25,34].

The Brillouin spectrum results from photons scattering caused by interaction with longitudinal acoustic phonons in the specimen. It features a central narrow Rayleigh peak, produced by photons scattered at the same frequency as the incident light, and two broader Brillouin peaks, due to photons scattered at lower (Stokes) and higher (Anti-Stokes) frequencies than the incident radiation[35]. These Brillouin peaks are symmetric, located in the GHz range, centered on the Brillouin frequency shift ($v_B$, red dotted lines in Fig. 1A) and share the same linewidth ($\Gamma_B$, blue arrows in Fig. 1A). In a backscattering geometry, $v_B$ and $\Gamma_B$ dependencies from $M'$ and $M''$ are shown in Eq. 1, where photon wavelength is λ, sample's index of refraction is $n$,

its density is $\rho$ and its longitudinal viscosity is $\eta_L$[35]:

$$v_B = \sqrt{\frac{M'}{\rho}} * \frac{2n}{\lambda}; \quad \Gamma_B = \frac{M''}{\rho v_B}\left(\frac{2n}{\lambda}\right)^2 = \frac{8\pi}{\rho}\left(\frac{n}{\lambda}\right)^2 \eta_L \qquad (1)$$

As will be explained further, $M'$ can be directly calculated from $v_B$ of the Brillouin spectra, whereas $\eta_L$ cannot be directly derived from $\Gamma_B$. The Brillouin spectra of our samples were acquired with our custom-built spontaneous Brillouin microscope (BM)[36], which consisted of an inverted standard optical microscope coupled to a double virtually imaged phased array (VIPA)-based spectrometer. Through the transmission of the VIPAs, the Brillouin triplet is repeated every 15 GHz, equal to the free spectral range (FSR). Our experimental Brillouin spectra (sketched in Fig. 1A, right panel and represented in Fig. 1B, lower panel, as dots) consisted of 2 Brillouin peaks, a Stokes and an Anti-Stokes, each belonging to a different dispersion order. The use of a spectrometer with a finite point spread function (PSF) introduces an instrumental broadening of the data, as explained by Eq. 2:

$$f_{\text{experimental}}(x) = f_B(x) \star f_{\text{PSF}}(x) \qquad (2)$$

Where $f_{\text{experimental}}(x)$ is the function describing the experimental data, $f_B(x)$ is the function describing the Brillouin lineshape, $f_{\text{PSF}}(x)$ is the function describing the PSF of the instrument and $\star$ denotes the convolution. If $f_{\text{PSF}}(x)$ is symmetrical, the convolution operation affects only the widths, not the peaks position: consequently, the relationship between $\Gamma_B$ (i.e., Brillouin intrinsic linewidth) and $\Gamma_{\text{experimental}}$ (i.e., Brillouin experimental spectra linewidth) depends on $f_B(x)$ and $f_{\text{PSF}}(x)$ functions[37] and in our case can be:

$$a. Linear: \quad \Gamma_{\text{experimental}} = \Gamma_B + \Gamma_{\text{PSF}} \qquad (2a)$$

if both $f_B(x)$ and $f_{\text{PSF}}(x)$ are Lorentzian functions;

$$b. Root\ mean\ square: \quad \Gamma_{\text{experimental}} = \sqrt{\Gamma_B^2 + \Gamma_{\text{PSF}}^2} \qquad (2b)$$

if $f_B(x)$ and $f_{\text{PSF}}(x)$ are Gaussian functions;

$$c. Voigt's\ profile: \quad \Gamma_{\text{experimental}} = \sqrt{\left(0.87 * \left(\frac{\Gamma_B}{2}\right)^2 + \Gamma_{\text{PSF}}^2\right) + 1.0692 * \left(\frac{\Gamma_B}{2}\right)} \qquad (2c)$$

if $f_B(x)$ is a Lorentzian and $f_{\text{PSF}}(x)$ a Gaussian, or vice versa

$$\Gamma_{\text{experimental}} = \sqrt{\left(0.87 * \left(\frac{\Gamma_{PSF}}{2}\right)^2 + \Gamma_B^2\right) + 1.0692 * \left(\frac{\Gamma_{PSF}}{2}\right)} \qquad (2c')$$

if $f_B(x)$ is a Gaussian and $f_{\text{PSF}}(x)$ a Lorentzian[37–39].

Hence, while sample $M'$ can be directly calculated from $v_B$ of the experimental data through Eq. 1, accurate estimation of $\eta_L$ from $\Gamma_B$ requires a deconvolution of the spectra that involves the knowledge of instrumental $\Gamma_{\text{PSF}}$. In a double-VIPA configuration, however, Rayleigh peaks are always suppressed (shaded areas of Fig. 1A, right panel): Brillouin spectra are thus typically subjected to an unknown value of $\Gamma_{\text{PSF}}$ and an unknown functional description of $f_{\text{PSF}}(x)$, that may be spectrometer-dependent (Fig. 1A, right panel). Moreover, $\Gamma_{\text{PSF}}$ can vary from day to day due to factors such as room temperature fluctuations, laser and VIPAs drifting[33,40], and differences in spectrometer alignment between users or across different sessions. Since alignment often requires repositioning the VIPAs, a deconvolution approach should be used that does not rely on a fixed value of $\Gamma_{\text{PSF}}$.

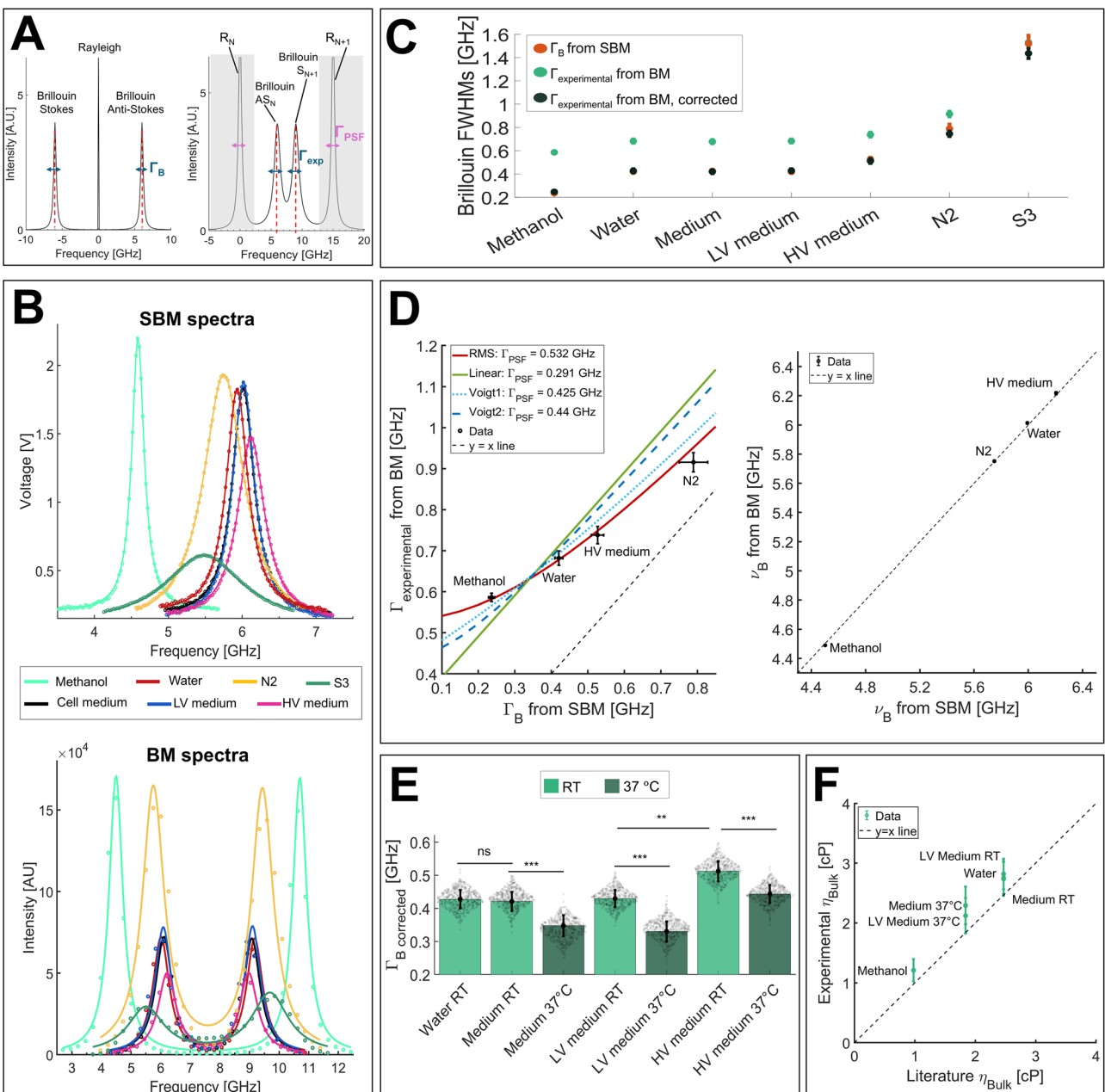

To overcome this problem and obtain the $\Gamma_{PSF}$ of our Brillouin microscope, we evaluated the relationship between $\Gamma_B$ and $\Gamma_{experimental}$ on different liquids and fitted the resulting curve using all the possible dependences (Eqs. 2a, b, c, c'). The precision of our BM was 9 MHz for the Brillouin shift and 18 MHz for the Brillouin FWHM (Supplementary Fig. 1). For every material, $\Gamma_{experimental}$ was obtained with our BM, while $\Gamma_B$ with a stimulated Brillouin microscope (SBM), i.e. a Brillouin microscope unaffected by spectral dispersive elements that allows access to the intrinsic linewidth of the materials[41]. The laser source of SBM operated at 780 nm, while BM worked at 660 nm; to ensure consistency, all results were converted to the reference wavelength of BM with a scaling factor (see "Methods"). The materials used for this step were: methanol, distilled water, regular cell medium, a low viscosity (LV) medium (obtained by mixing regular cell culture medium with 6 kDa Dextran), a high viscosity (HV) medium (obtained by mixing culture medium with 500 kDa Dextran), N2 and S3 (two viscosity standards, see Methods). Their viscosities spanned across a wide range (Supplementary Table 1, Supplementary Table 2). Figure 1B upper panel shows their raw Brillouin spectra obtained with SBM

(dots) and the relative fits (continuous lines; here, we acquired only the Anti-Stokes peak), while Fig. 1B lower panel shows their raw Brillouin spectra acquired with our BM (dots) and their fits (solid lines). From the fits of the spectra, we determined the Brillouin widths of the liquids using SBM and BM at 660 nm: in particular, Fig. 1C shows $\Gamma_{experimental}$ (i.e., widths obtained with BM, subjected to VIPAs enlargement, green dots), and $\Gamma_B$ (i.e., widths obtained with SBM, not affected by any form of dispersion, red dots). Here, it is clear that below the linewidth value of the S3 liquid, the instrumental width of the BM leads to artefactually higher values compared to SBM. At S3 point, SBM and BM widths were identical. Moreover, water, regular cell medium and LV medium had the same widths in BM and SBM and thus their values were redundant.

Consequently, as viscosity calibrators, we used only four materials: methanol, water, HV medium and N2. We show their Brillouin widths (Fig. 1D, left) and shifts (Fig. 1D, right), both converted to the wavelength of the BM. From the first graph, we determined $\Gamma_{PSF}$ by fitting $\Gamma_{experimental}$ against $\Gamma_B$: the fittings included linear (green solid line, described by Eq. 2a), root mean square (RMS, red solid line, Eq. 2b) and Voigt profiles (dotted

**Fig. 1 | Brillouin full widths at half maximum (FWHMs) obtained from different materials for viscosity calibration.** **A** left The Brillouin spectrum of a material is composed of a central narrow Rayleigh peak (due to photons scattered at the same frequency of the incident radiation) and two broader, identical Brillouin peaks (Stokes and Anti-Stokes), whose center is the Brillouin shift $v_B$ (red dotted lines) and width is $\Gamma_B$ (blue arrows). **A** right Brillouin triplet of panel **A** imaged with a VIPA spectrometer of free spectral range (FSR) = 15 GHz. The spectrometer repeats the Brillouin triplet every 15 GHz and introduces an instrumental broadening due to its finite point spread function (PSF): here, the Rayleigh width is $\Gamma_{PSF}$ (purple arrows) and the Brillouin width is $\Gamma_{experimental} > \Gamma_B$. During data acquisition with a double-VIPA setup, only the Stokes (S) and Anti-Stokes (AS) Brillouin peaks of two adjacent diffraction orders ($N$ and $N+1$) are recorded, while their respective Rayleigh peaks ($R_N$ and $R_{N+1}$) are suppressed (shaded areas). **B** top Brillouin spectra (only Anti-Stokes peak) obtained with SBM (dots) and the corresponding fits (continuous lines), from which we extrapolated $\Gamma_B$, i.e., the width unaffected by instrumental broadening. SBM laser source operated at 780 nm; to facilitate comparison with BM results, the spectra are here shown to an equivalent wavelength of 660 nm (described in the "Methods"). **B** bottom Brillouin spectra obtained with BM (dots) and the corresponding fits (solid lines), from which we extrapolated $\Gamma_{experimental}$, i.e., the width affected by VIPAs broadening. The BM laser source was 660 nm: we used this wavelength as a reference for all the measurements. **C** Corresponding $\Gamma_B$ (red dots) and $\Gamma_{experimental}$ (green dots) values for the different materials. Water, medium and LV medium all resulted in the same widths, and their values were redundant. Black dots are corrected $\Gamma_{experimental}$ and obtained once properly deconvolved with a Gaussian $\Gamma_{PSF}$. **D** left $\Gamma_{experimental}$ vs $\Gamma_B$ for the four calibration materials, together with the different fits: linear ($y = x + \Gamma_{PSF}$: green line, yielding $\Gamma_{PSF} = 0.291$ GHz), root mean square (RMS: $y = \sqrt{x^2 + \Gamma_{PSF}^2}$: red line, $\Gamma_{PSF} = 0.532$ GHz) and Voigt

(Voigt1: $y = \sqrt{\left(0.87 * \left(\frac{x}{2}\right)^2 + \Gamma_{PSF}^2\right)} + 1.0692 * \left(\frac{x}{2}\right)$, in which PSF is a gaussian and Brillouin is a Lorentzian: cyan dotted line, $\Gamma_{PSF} = 0.425$ GHz; Voigt2:

$y = \sqrt{\left(0.87 * \left(\frac{\Gamma_{PSF}}{2}\right)^2 + x^2\right)} + 1.0692 * \left(\frac{\Gamma_{PSF}}{2}\right)$, in which the roles were reversed:

blue dashed line, yielding $\Gamma_{PSF} = 0.440$ GHz). The Gaussian fit resulted in a more significant description of the data. Black dashed line: $y = x$ line, showing that $\Gamma_{experimental} > \Gamma_B$ always. Data are shown as mean ± SD over $n = 500$ repeated measurements for BM and $n = 100$ for SBM. SBM data are shown in Supplementary Table 1. **D** right the Brillouin shifts of the calibration materials acquired with the SBM and BM matched, showing that the convolution operation does not affect Brillouin peak position but only its width. **E** Deconvolved widths of different liquids at room temperature (RT), heated at 37 °C or having different concentrations of Dextran, show that Brillouin linewidth is extremely sensitive to temperature (medium RT vs 37 °C: $p = 0.003$; LV medium RT vs 37 °C: $p = 0.005$; HV medium RT vs 37 °C: $p = 0.001$) and polymer concentration (water vs medium RT: $p = 0.85$; LV medium RT vs HV medium RT: $p = 0.008$). Medium: regular cell medium; LV medium: low viscosity medium, corresponding to culture cell medium with 6 kDa Dextran; HV medium: high viscosity medium, corresponding to cell medium with 500 kDa Dextran. **F** $\eta_{bulk}$ values, calculated from deconvolved widths of each material (y-axis), agree with theoretical values found in literature (x-axis). Dotted line: $y = x$ line, showing a good correlation between experimental and theoretical data. Data are shown as mean ± SD, performed over $n = 500$ measurements. Statistical analysis has been performed using ordinary one-way ANOVA. **$p < 0.01$; ***$p < 0.005$; ns not significative.

lines, Eqs. 2c and 2c': cyan for a Gaussian–Lorentzian mixture with a Gaussian PSF and Lorentzian Brillouin component, and blue for a Lorentzian–Gaussian mixture with the roles reversed[38,39]). The best agreement between theoretical and experimental widths was given by the RMS behavior, yielding $\Gamma_{PSF} = 0.532$ GHz, whereas the Voigt2 and the linear models provided less accurate representations, suggesting that the instrumental PSF was unlikely to be a pure Lorentzian. To validate these findings, we compared $\Gamma_{PSF}$ with the width obtained by fitting the Rayleigh peaks with a Gaussian or a Lorentzian: the former fit showed superior accuracy in reproducing the instrumental line shape, particularly at the tails of the signal (Supplementary Fig. 2A). Nevertheless, both fits yielded similar half-maximum values, resulting in comparable widths (Supplementary Fig. 2B), which corresponded to the $\Gamma_{PSF}$ found from root-square fitting of the calibration materials widths (Supplementary Fig. 2C). These results confirmed that the behavior between $\Gamma_{experimental}$ vs $\Gamma_B$ was not linear nor Voigt and that the VIPAs were mostly described by a Gaussian. Moreover, Fig. 1D right, shows that the Brillouin shifts of calibration materials were equal in SBM and BM: this confirmed that the convolution operation of Eq. 2 affects only the widths and not the positions of the peaks. Furthermore, while the Brillouin shifts of the calibration materials remained consistent across different measurement days (Supplementary Fig. 3A), their linewidths varied significantly due to the sensitivity of $\Gamma_{PSF}$ to daily spectrometer alignment, fluctuating by hundreds of MHz (Supplementary Fig. 3B) due to room temperature fluctuations, confirming the need for a non-constant value of $\Gamma_{PSF}$. Despite this, $\Gamma_{experimental}$ vs $\Gamma_B$ relationship consistently followed a root-mean-square dependence, and the determined $\Gamma_{PSF}$ allowed for effective data deconvolution, ensuring reliable viscosity values corrected across different days.

Through the retrieved $\Gamma_{PSF}$, we could correct for instrumental broadening by inverting the root-mean-square dependence and thus finding $\Gamma_{B\ corrected}$:

$$\Gamma_{B\ corrected} = \sqrt{\left(\Gamma_{experimental}^2 - \Gamma_{PSF}^2\right)} \quad (3)$$

As shown by the black points of Fig. 1C, $\Gamma_{B\ corrected}$ matched $\Gamma_B$ values obtained with SBM confirm the goodness of this method in correcting experimental Brillouin widths obtained with a BM.

Our data confirm that BM is an instrument capable of measuring reliable FWHMs once properly deconvolved via the instrument PSF. Its sensitivity to subtle differences in materials' viscosities $\eta_L$ is evident in Fig. 1E, where variations in temperature and polymer concentrations led to discernible changes in Brillouin FWHM values: the same medium held at room temperature (RT) or at 37 °C showed significantly different Brillouin widths, and increasing concentration of Dextran led to significantly different FWHMs, both at RT and at 37 °C. Regular cell medium and LV medium showed the same FWHMs.

From hydrostatic calculations in pure liquids[35], it is known that longitudinal viscosity $\eta_L$ can be well approximated by the sum of shear viscosity ($\eta_{shear}$) and bulk viscosity ($\eta_{bulk}$), as detailed in Eq. 4[42]:

$$\Gamma_B = \frac{8\pi}{\rho}\left(\frac{n}{\lambda}\right)^2 \eta_L \sim \frac{8\pi}{\rho}\left(\frac{n}{\lambda}\right)^2 \left(\frac{4}{3}\eta_{shear} + \eta_{bulk}\right) \quad (4)$$

To check if deconvolved $\Gamma_B$ from experimental Brillouin data were reliable values, we measured $\rho$, $n$ and $\eta_{shear}$ for every sample (Supplementary Table 2) and plotted retrieved $\eta_{bulk}$ values (extracted from Eq. 4) in Fig. 1F. The comparison of our experimentally retrieved values with known values extracted from literature[42–45] was within two standard deviations of difference from the theoretical values; $\eta_{bulk}$ significantly varied with temperature, as expected[35].

Overall, this confirms that FWHM values fitted from Brillouin spectra, once properly deconvolved for instrumental broadening by comparing widths of calibration materials acquired with BM and SBM (Fig. 1D), can indeed be used to retrieve reliable viscosity values of a sample. By just taking the Brillouin spectra of 2 extra liquids (HV media and N2) together with the already established calibration materials (water and methanol, necessary for pixel-to-GHz transformation[36]), it is possible to completely define the Longitudinal Modulus and Viscosity of a sample that compensates for day-to-day instrument calibration oscillations. We propose this as a future method to calibrate Brillouin microscopes for proper FWHM calculation.

### Alterations of cellular longitudinal modulus and viscosity in differently viscous culture media

We then characterized if extracellular viscosity could act as a trigger that modifies cellular mechanical properties (i.e., longitudinal modulus and

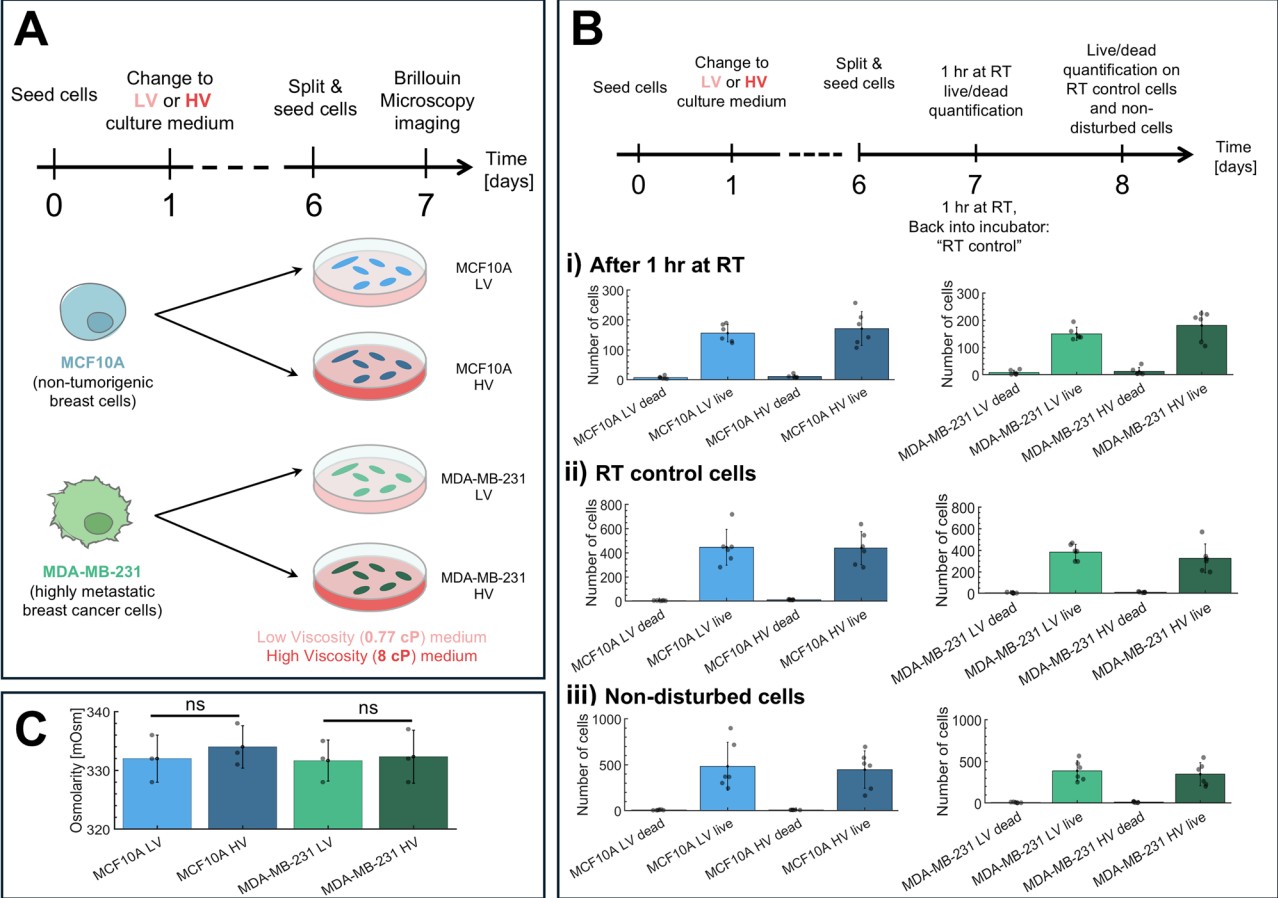

**Fig. 2 | Low and high media viscosity experimental setup on cell lines. A** Timeline of cell seeding and Brillouin measurement procedure. Cells (MCF10A as non-tumorigenic breast cells, in blue, and MDA-MB231 cells as highly metastatic, in green) were seeded on day 0, and media were changed on the following day to either LV media (light pink) or HV media (dark pink); after 1 week from seeding, we performed Brillouin Microscopy imaging. **B** Timeline of cell seeding and live/dead quantification. Cells on day 6 were split and seeded into three groups: 1 h at room temperature (RT), RT control, and non-disturbed. The 1 h RT group is left out at RT for 1 h to simulate 1 h of Brillouin acquisition. RT control involves leaving cells out at RT for 1 h and returning cells to 37 °C, serving as a control to the 1 h RT cells. Non-disturbed cells are incubated at 37 °C for 24 h to serve as a control for the two previous conditions. (i) Number of live vs dead cells after 1 h incubation at RT for each respective cell line. (ii) Number of live vs dead cells for RT control cells. (iii) Number of live vs dead cells for non-disturbed cells. **C** Osmolarity remains unchanged across LV and HV medias. Data are shown as mean ± SD performed over $n = 6$ (panel **B**) or $n = 3$ (panel **C**) independent measurements. ns not significant (t-test).

viscosity) by exploiting our BM. Specifically, we aimed to determine if cells cultured in media with varying viscosities could adapt their nuclear mechanical properties in a way similar to how cells grown on hard or soft substrates adjust their nuclear stiffness in response to the external environment[11,46–48].

The human breast cells used for this purpose were MCF10A, as a non-tumorigenic breast cell line and MDA-MB-231 as a highly metastatic cell line[24]. Cells were seeded and grown in differently viscous culture media (LV as low viscosity medium or HV as high viscosity medium, with viscosities equal to 0.77 and 8 cP) for one week, mirroring long-term measurements already described[21] (Fig. 2A). To ensure the viability of cells in the HV media, we performed live/dead assays. To simulate Brillouin measurement conditions, we let cells incubate at room temperature (RT) for 1 h and then performed the live/dead analysis (Fig. 2Bi, Supplementary Fig. 4A). As a control, we let cells incubate at RT for 1 h before returning them to 37 °C for an additional 24 h of incubation (Fig. 2Bii). An additional control group of non-disturbed cells was incubated at 37 °C for 48 h post-seeding on the 6th day (Fig. 2Biii). All three conditions were found to have less than 7% cell dead cells. Morphologies such as cellular area, circularity, and aspect ratio were also analyzed (Supplementary Fig. 4B). Notably, the averaged cellular area of MDA-MB-231 cells in HV media was larger compared to MDA-MB-231 cells in LV media and was comparable to the averaged cellular area of MCF10A cells in both LV and HV media. These results show that cells were viable and proliferating in HV media. Upon introducing polymers to modify the viscosity of the culture medium from 0.77 cP to 8 cP, we ensured that osmolarity remained unchanged (Fig. 2C)[49].

Following the long-term treatment, we imaged live MCF10A and MDA-MB-231 cells in both LV and HV medium with our custom-made Brillouin Microscope. Here, a standard brightfield transmission unit and a fluorescence module provided morphological insights of the cells and fluorescent recognition of HOECHST-stained nuclei; nuclear mechanical properties were quantified by exploiting Brillouin microscopy as a contact-less and non-invasive methodology to retrieve Brillouin spectra point-by-point, in three dimensions and with sub-micrometric spatial resolution[26]. Experimental outcomes of brightfield, fluorescence and Brillouin maps obtained on MCF10A and MDA-MB-231 are depicted in Fig. 3A: here, corrected Brillouin FWHMs were obtained point-by-point by utilizing the $\Gamma_{PSF}$ obtained prior to experimentation (through Eq. 3).

Data in Fig. 3 show that cells respond to higher extracellular viscosity by reducing their nuclear longitudinal viscosity, as evident when transitioning from LV to HV culture media for both non-cancerous MCF10A and cancer MDA-MB-231 cells. Interestingly, cancer cells always exhibited lower viscosity with respect to non-cancer cells, both in LV and HV media. The longitudinal modulus acted independently from viscosity: in response to higher extracellular viscosity, only cancer cells tended to augment nuclear modulus, while non-cancer cells were unaffected by environmental changes.

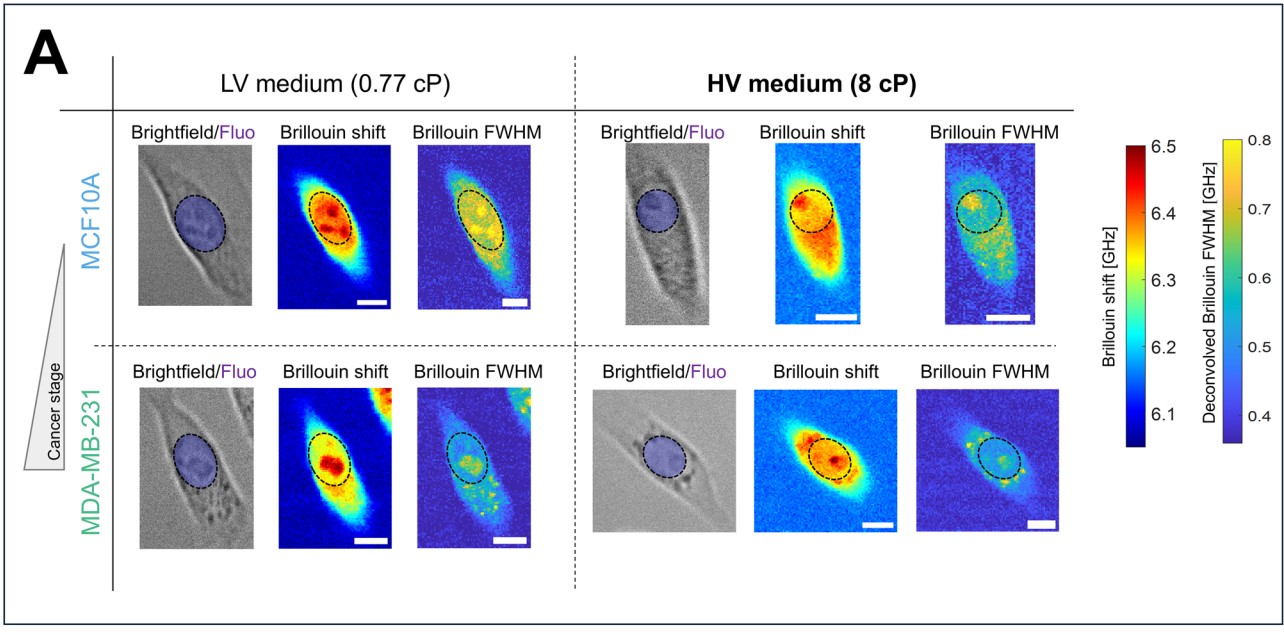

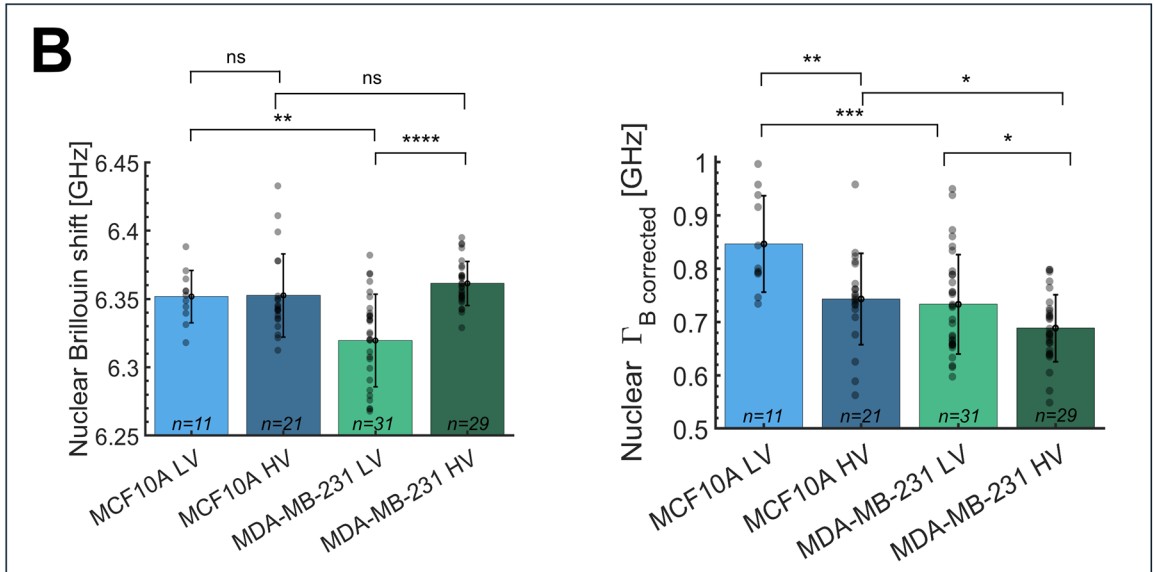

**Fig. 3 | Brillouin microscopy maps obtained with our custom-made BM.**
**A** Representative brightfield, Brillouin shift (jet colormap) and Deconvolved Brillouin FWHMs (parula colormap) maps of MCF10 and MDA-MB-231 cells grown for 1 week in media having different viscosities (LV: low viscosity medium, 0.77 cP; HV: high viscosity medium, 8 cP). See the timeline of the experiment in Fig. 2A. The dotted circles represent the nucleus position, extracted from HOECHST fluorescence maps, over which we averaged Brillouin shifts and widths. The Brillouin FWHM maps have been deconvolved point by point by considering the PSF width obtained prior to experimentation, as in Eq. 3. Scale bars = 10 microns. **B** Bar plots of nuclear Brillouin shifts and Deconvolved FWHMs under different media viscosities and cell lines. Data in the bar plots are shown as mean ± SD, derived from $N = 5$ independent measurements; the number of total data points n is written in every bar

plot. Regarding Brillouin shift changes (left panel), under normal conditions of LV medium, MDA-MB231 had lower modulus than MCF10A ($p = 0.003$), while at HV their values matched ($p = 0.45$); when comparing LV with HV conditions, only cancer cells tended to augment their nuclear modulus ($p < 0.0001$), while non-cancer ones were unaffected ($p = 0.94$). Regarding corrected Brillouin FWHMs (right panel), MDA-MB231 exhibited lower longitudinal viscosity with respect to MCF10A in both LV medium ($p = 0.0008$) and in HV medium ($p = 0.047$); when passing from LV to HV medium, instead, both non-cancer ($p = 0.0036$) and cancerous ($p = 0.047$) cells reduced their nuclear longitudinal viscosity. Statistical analysis has been performed using ordinary one-way ANOVA using a Sidak correction for multiple comparisons. *$p < 0.05$; **$p < 0.005$; ***$p < 0.001$; ****$p < 0.0001$; ns not significant.

Under normal conditions (LV), MDA-MB-231 had a lower modulus than MCF10A, while at high viscosity media, cancer cells increased their nuclear modulus to equal the value of non-cancerous cells in the same conditions.

## Discussion

A Brillouin spectrum allows to measure the longitudinal modulus $M'$ and viscosity $\eta_L$ of any material, once its index of refraction $n$ and density $\rho$ are known (Eq. 1; Fig. 1). When imaging Brillouin spectra with a spectrometer,

$M'$ is straightforwardly obtained from the Brillouin shift $\nu_B$, while the full-width at half maximum $\Gamma_B$ is affected by the instrumental broadening and thus cannot be directly related to $\eta_L$. Hence, Brillouin spectra acquired with our custom-built Brillouin microscope (BM)[36], equipped with a double VIPA spectrometer and working in a back-scattering configuration, had to be deconvolved through Eq. 3 to retrieve intrinsic Brillouin linewidths. The role of viscosity in Brillouin microscopy remains relatively underexplored in the current literature, where many only report the Brillouin frequency shift

corresponding to the elastic or "storage" modulus. While some do co-report both Brillouin frequency shift and Brillouin linewidth[50,51], they do not perform spectral deconvolution, which is a necessary step to quantify the intrinsic linewidth associated with viscous properties. Accurate estimation of the Brillouin linewidth using VIPA-based spectrometers has indeed been challenging[27–29]. While techniques such as impulsive stimulated Brillouin scattering and time-domain approaches have been developed to achieve higher spectral resolution and more accurate Brillouin linewidth measurements, their applicability to living biological samples remains limited[30,31,52–54]. In light of this, we sought to develop a generalizable and straightforward method to deconvolve the Brillouin spectra acquired with spontaneous Brillouin microscopes by requiring only the addition of two extra standards to the calibration protocol established in literature[33,36].

To achieve this, we acquired data of materials having viscosities spanning across a wide range of values with our BM and with a stimulated Brillouin microscope (SBM); the SBM, being free from spectral dispersive elements, enables the direct measurement of the true Brillouin linewidth of the materials[41]. In such a manner, we used linewidth data obtained from SBM solely to deconvolve the spectra acquired with our BM on cells. Importantly, SBM was not applied to biological samples, as its use in this context remains limited due to the high laser powers involved, which can cause thermal damage to cells[32,55,56]. In contrast, spontaneous Brillouin microscopy is currently the standard technique for probing cellular and tissue biomechanics[57]; it is more widely adopted, less expensive, and better suited for biological applications than SBM.

We measured the widths of methanol, water, cell culture medium, low viscosity (0.77 cP) medium, high viscosity (8 cP) medium, N2 and S3 oils (two viscosity standards). Regular cell and LV media widths matched the water one, as expected (cell medium is known to have the same viscosity as water, and 6 kDa Dextran should not increase the viscosity of the medium[21]), while S3 liquid was not affected by the instrumental broadening. We thus decided to only acquire non-redundant data as calibration materials for spectra deconvolution, i.e., methanol, water, high viscosity cell medium and N2 oil. From the fit of these spectra (Fig. 1B), we evaluated the relationship between $\Gamma_B$ (acquired with the SBM, not affected by any form of dispersion) and $\Gamma_{experimental}$ (acquired with the BM, subjected to VIPAs enlargement) of the calibration materials; here, $\Gamma_{experimental} > \Gamma_B$ always, as evident from Fig. 1C and the black dotted y = x line in Fig. 1D, left panel.

Theoretically, Brillouin scattering is well described by a damped harmonic oscillator or Lorentzian distribution[35], while the unknown PSF due to VIPAs contribution could exhibit either Lorentzian or Gaussian characteristics, or a combination of the two, i.e., a Voigt or a pseudo-Voigt profile[58]. Thus, to fit the $\Gamma_{experimental}$ vs $\Gamma_B$ data we used different models: linear (in which both functions are described by Lorentzian, Eq. 2a), root mean square (in which both functions are Gaussian, Eq. 2b) and Voigts (Eqs. 2c and 2c': Voigt1, a mixture of a Gaussian PSF and Lorentzian Brillouin component, or Voigt2, where the roles were inverted[38,39]). The Voigt1 model demonstrated a good alignment, indicating that the unknown elastic line could effectively be represented by a Gaussian distribution; the root-mean square dependence best described the relationships between $\Gamma_B$ and $\Gamma_{experimental}$, resulting in an instrumental broadening ($\Gamma_{PSF}$) oscillating between 530 and 730 MHz (Fig. 1D, Supplementary Fig. 3B), typical of a double-VIPA setup[59,60] and subjected to daily spectrometer alignments. To validate these findings, we compared $\Gamma_{PSF}$ with the width obtained by a longer and less accurate procedure, i.e. fitting the Rayleigh peaks with a Gaussian or a Lorentzian (Supplementary Fig. 2A). Here, elastic peaks were imaged on the camera by lowering the laser intensity to the minimum and opening the slits of the spectrometer, thus allowing for Rayleigh signals to pass and be captured by the spectrometer camera. Gaussian function was more accurate in reproducing the instrumental line shape, confirming the results found from calibration materials fitting; both Lorentzian and Gaussian fits, however, resulted in comparable widths (Supplementary Fig. 2B) that matched the $\Gamma_{PSF}$ found from root-square fitting of the calibration materials (Supplementary Fig. 2C). These results confirmed that the relationship between $\Gamma_{experimental}$ and $\Gamma_B$ was not linear nor Voigt, but

Gaussian and that the VIPAs response function was mostly described by a Gaussian distribution. Of note, Voigt2 and the linear models provided much less accurate representations, suggesting that the instrumental PSF was unlikely to be a pure Lorentzian and that, in general, the deconvolution of Brillouin widths with a simple subtraction $\Gamma_{PSF}$ might induce several errors. Other works in literature show a deconvolution of Brillouin spectra by subtracting a fixed $\Gamma_{PSF}$ value from Brillouin widths[59,61], assuming a Lorentzian relationship between $\Gamma_{experimental}$ and $\Gamma_B$ for their spectrometer, but our results show that this procedure might result in a severe underestimation of the retrieved $\Gamma_B$ if the comparison with theoretical values is not performed. Additionally, using a fixed spectrometer linewidth for deconvolution may overlook day-to-day variability due to changes in system alignment, a factor that should be accounted for. Other studies, furthermore, present valuable insights into Brillouin linewidths, but do not perform any deconvolution on the spectra[50,51]: as recently highlighted[33], this leads to systematic overestimation of linewidths, failing to correct for instrumental broadening and thermal drifts that occur between measurement sessions.

Through the retrieved $\Gamma_{PSF}$, we corrected for instrumental broadening, thus finding $\Gamma_{B\ corrected}$ via Eq. 3: these values matched $\Gamma_B$ obtained from SBM (black points of Fig. 1C), confirming the goodness of this method in correcting experimental Brillouin linewidths obtained with a BM. We checked that instrument broadening did not affect the Brillouin shifts, as the shifts of calibration materials acquired with SBM matched the BM (Fig. 1D, right panel) and were very consistent across different measurements days (Supplementary Fig. 3A). It is important to note that the use of a high numerical aperture objective also broadens and shifts the Brillouin peak in both BM and SBM; however, in a backscattering configuration like ours, this effect is known to be minimal with respect to other scattering angles, that would induce a higher broadening[62]. To avoid this influence, we acquired Brillouin spectra in both BM and SBM using the same objective and in the same backscattering configuration, thereby correcting BM widths only for the spectrometer presence.

Finally, variations in temperature (room temperature vs 37 °C) and polymer concentrations (6 kDa Dextran vs 500 kDa Dextran) led to discernible changes in FWHM values (Fig. 1E)[63]. We then calculated $\eta_{bulk}$ (Eq. 4, where we did not take into account other thermodynamical parameters whose contribution to $\eta_L$ is negligible in many liquids[35,64]) from $\Gamma_{B\ corrected}$, $\eta_{shear}$, $n$ and $\rho$ of every sample (Fig. 1F). Importantly, we could put together $\eta_{shear}$ (measured with a viscometer at low frequencies) with $\eta_L$ (obtained from Brillouin data taken at GHz frequencies) because water[65], methanol and even Dextran (dispersed at low concentrations[66,67]) are all Newtonian fluids: thus, their viscosities are independent from shear rate. The difference between our experimentally retrieved $\eta_{bulk}$ values was within two standard deviations from values found in literature[43–45]. Here, an offset from the diagonal was observed, likely arising from a slight overestimation of $\Gamma_B$, but other terms such as n or $\eta_{shear}$—experimentally retrieved with different techniques—might also contribute; additionally, the absence of error bars in the literature values could influence the observed deviation on the x-axis. In any case, the obtained bulk viscosity varied significantly with temperature, as expected[43]. $\eta_{bulk}$ of other used liquids (as N2, S3, or HV medium) is not available in literature, so their comparison was not possible.

In such a manner, we had a benchmark to validate the goodness of the method to extrapolate the intrinsic Brillouin linewidths, crucial to have information about the longitudinal viscosity of a sample. Notably, this deconvolution approach is designed to be broadly applicable to different Brillouin Microscopes. By measuring the Brillouin spectra of the four calibration materials, users can determine the relationship between their experimental linewidths and our intrinsic values obtained with the SBM (Supplementary Table 1), enabling spectral deconvolution specific to their system. Importantly, this method is not dependent on the specific optical design of the spectrometer nor on the wavelength of the laser source. The only critical parameter is the NA of the objective, which should match the one used in our study (NA = 0.95): a different NA may introduce spectral broadening and shifting, potentially affecting the reliability of the deconvolution[62].

We then acquired Brillouin Microscopy data on human breast cells lines (MCF10A as a non-tumorigenic breast cell line and MDA-MB-231 as a highly metastatic cell line[24]) cultured in media of different viscosities to investigate: (i) whether breast cells at different cancer stages exhibit differences in longitudinal viscosity, considering their known differences in stiffness[1,5–8]; (ii) if variations in extracellular media viscosity may induce changes in the mechanical properties of the cells; (iii) if these variations were contingent upon cells cancer stage. Existing literature reports that cells grown on hard or soft substrates modify their own nuclear stiffness[11,46–48]; therefore, we wanted to investigate whether viscosity could have a similar effect.

There are many agents that can be used to increase the viscosity of cellular media, with the most popular ones being methylcellulose[68,69], dextran[70], and polyvinylpyrrolidone[71]. We first experimented using 6 kDa and 500 kDa dextran to make 0.77 cP and 8 cP cell media, respectively. However, in our experience, both MCF10A and MDA-MB-231 cells reacted poorly to 500 kDa dextran, and the majority of cells would die after a minimum of 24-h exposure, making dextran a non-ideal media thickening agent for long-term exposure. We next tried using polyvinylpyrrolidone (PVP) as demonstrated by Bera et al.[21], and found that HV media made using PVP were much more tolerated by the cells and thus were used in all experiments that required HV media in living cells. To show that the cells were proliferative and viable in HV media made using PVP, we performed live/dead assays on cells after long term incubation mimicking Brillouin acquisition experimental procedures (Fig. 2B) after (i) 1 h of RT incubation, (ii) 24 h post 1 h of RT incubation to serve as a control for i, and (iii) 48 h post seeding cells without disturbance (i.e., noninterrupted incubation at 37 °C). In all three conditions, we found cell death to be less than 7% (Fig. 2Bi–iii, Supplementary Fig. 4A) and that there was no significant difference between the number of live cells between conditions ii and iii, demonstrating cellular proliferation and viability in HV media made using PVP. We also analyzed cellular area, circularity and aspect ratio (Supplementary Fig. 4B): from LV to HV, MDA-MB-231 cells were larger, comparable to the cellular area of MCF10A in LV and HV medias. Additionally, we demonstrated that the osmolarity of cell culture medias remained unchanged with and without PVP (Fig. 2C): thus, alterations in Brillouin shift and FWHMs solely reflected changes in the mechanical properties of the sample, unaffected by altered osmolarity of the buffer[49].

Our in vitro Brillouin Microscopy data (Fig. 3) show that in response to elevated extracellular fluid viscosity, cells exhibited a tendency to reduce their nuclear longitudinal viscosity. This trend was observed when transitioning from LV (0.77 cP) to HV (8 cP) medium for both non-cancerous cells MCF10A and cancer cells MDA-MB-231. This aligns with recent research[21,22,72] where increased external viscosity correlated with enhanced cellular velocity and motility in cancerous cells (as MDA-MB-231) and other non-cancerous cell lines. This result is interesting, as cellular motility is dependent on intracellular movements of cytosolic components and subsequently nuclear components[73]; thus, if a cell's contents are less viscous, they will require less energy to migrate[74]. Moreover, a decrease in nuclear viscosity suggests that the nucleus may become more deformable and less resistant to mechanical stresses. This is particularly relevant in the context of tumor progression: our results on nuclear viscosity thus might provide an important biomechanical motivation to the enhanced motility seen in cancer cells in highly viscous extracellular fluids[21,22,72]. Indeed, metastatic cells migrate through crowded and confined microenvironments, such as the extracellular matrix and endothelial barriers during intravasation and extravasation[1,2,7,15,73]. The nucleus is the largest and stiffest organelle in the cell, and its mechanical properties are the limiting factor for the deformability and migration of the whole cell through confined spaces[1,2,7,15,73]: reduced nuclear viscosity may thus facilitate nuclear deformation and enhance a cell's ability to squeeze through tight pores and vessel walls, thereby promoting invasive behavior and metastatic dissemination. Furthermore, decreased nuclear viscosity influences gene expression by activating different mechanotransduction pathways, as already shown[72]. Consequently, cells may adapt their internal mechanical properties in response to changes in external media viscosity, leading to higher deformability and to better hydrodynamic responses, thus potentially increasing cellular velocity.

Under normal conditions (LV), metastatic MDA-MB-231 cells had lower longitudinal modulus and viscosity than non-cancerous MCF10A cells, indicating a connection between breast cancer's metastatic propensity with lower stiffness and viscosity. While the former finding was well established in literature, the latter is novel and concurs with recent research on lower viscosities of MDA-MB-231 cells using magnetic rotational spectroscopy at low frequencies[75]. These findings are also consistent with other studies[50] reporting lower Brillouin linewidths in cells from more aggressive metastatic stages compared to early-stage cancer cells. This supports the potential use of Brillouin linewidth as a parameter for distinguishing tumor tissues based on metastatic potential. Notably, that study employed an objective with a significantly lower numerical aperture than ours, suggesting that the broadening induced in our data by the high NA does not compromise the validity of our deconvolution protocol.

In HV medium instead, metastatic MDA-MB-231 cells retained lower longitudinal viscosity than control, but modulated their nuclear modulus to equal the value of non-cancerous cells. The decrease in viscosity as a response to augmented external viscosity appears to be a universal behavior among cells, irrespective of their cancer propensity; cancer cells, however, also exhibited changes in longitudinal modulus, with HV MDA-MB-231 cells displaying increased modulus compared to LV conditions, while normal MCF10A cells maintained consistent modulus. In this context, it will be of interesting future work to deepen why cancer cells retain low viscosity while adapting their internal stiffness to the environment, consistent with previous observations[47].

All these results indicate a fascinating and dynamic behavior of longitudinal modulus and viscosity, which could serve as potential markers for discerning cell metastatic potential. To date, the role of viscosity in Brillouin Microscopy remains unclear and has not been explored much in the literature; our study further emphasizes the potential of deconvolved Brillouin width maps as reliable indicators of longitudinal viscosity in living cells. These maps also demonstrate the high spatial resolution of our Brillouin microscope, allowing us to resolve subcellular structures (visible in Brillouin shift and widths maps, as well as in brightfield images) such as the nucleoli in the nucleus[76,77] and localized features in the cytoplasm, already detected in other cell lines[78–80]. These cytoplasmic features, appearing as dark spots in brightfield images captured prior to Brillouin acquisition, confirm that they are not artifacts introduced by the Brillouin imaging process.

A potential limitation of this study is that our analysis was focused on the nucleus, which is the largest and stiffest organelle in the cell; it is therefore interesting from a mechanical standpoint, representing the limiting factor in cellular deformability. Moreover, our protocol for cellular Brillouin acquisitions is refined to give us the best representation of the nucleus of the cells. Extending this analysis to the cytoplasm could offer additional insights, especially considering its role in cell motility and spread. However, our Brillouin measurements in the cytoplasm were limited by high noise: this was due to its proximity to the growing substrate, where reflections from the glass surface produced strong Rayleigh scattering, interfering with spectral acquisition.

Another possible limitation is that we always assumed the presence of a single peak in the Brillouin spectra when analyzing cell data. Indeed, raw Brillouin spectra clearly showed a single sharp peak, indicating that the samples were homogeneous on the phonon wavelength scale (~250 nm with $\lambda = 660$ nm[81]): the assumption of a single peak thus appeared to be justified. Moreover, the Brillouin FWHM was always very sensitive to the quality of the spectra and to the model used to fit the data, much more than the Brillouin shift; to be consistent, we always fitted the spectra by using the same number of points and the same function (see Methods). Finally, in Eq. 4 we did not consider other thermodynamical parameters (namely thermal conductivity, heat capacity and adiabatic index) as their influence on $\eta_L$ is negligible in many liquids[35]; given the strong agreement between $\eta_{bulk}$ and our experimentally retrieved values, we can conclude that it was an appropriate decision.

In summary, employing our methodology to properly interpret the Brillouin line shape width as a reliable viscosity indicator, we showed the interplay between extracellular viscosity and cellular mechanical properties. Remarkably, cancer cells exhibited a propensity to decrease nuclear viscosity and increase nuclear modulus in response to heightened extracellular fluid viscosity, underscoring their adaptive capabilities to the local environment. Our study contributes to the growing body of evidence suggesting that viscosity, akin to stiffness, may serve as a potential biomarker for discerning cancer propensity. The observed changes in cellular mechanics shed light on the dynamic interplay between cells and their microenvironment, with implications for understanding cancer progression and metastasis.

## Methods

### Cell culture maintenance
MCf10A cells were a gift from Michele Vitolo from the University of Maryland School of Medicine. MCF10A cells were cultured in complete growth media consisting of DMEM/F12 (Invitrogen #11330-032), 5% Horse Serum (Invitrogen #16050-122), EGF (Peprotech), Hydrocortisone (Sigma #H-0888), Cholera Toxin (Sigma #C-8052), Insulin (Sigma #I-1882), Pen/Strep (Invitrogen #15070-063). MDA-MB-231 cells were purchased from ATCC (HTB-26) and cultured in complete growth media consisting of DMEM (ATCC 30-2002), 10% FBS (ATCC 30-2020), Pen/Strep (Invitrogen #15070-063). Cells were either maintained in low viscosity (LV) media or switched to high viscosity (HV) media 24 h after seeding and maintained in HV media for 7 days, passing when reaching ~70–80% confluency. Cells were resuspended and spun down using the respective LV media, as the HV media disrupted the pelleting of the cells. On the 6th day of culturing, cells were passaged and seeded into 35 mm ibidiTreat dishes (Ibidi) at low density to ensure single cell formation; the next day, cells were live imaged at the Brillouin Microscope. Cells were not checked for mycoplasma contamination, nor were they authenticated.

### Low and high viscosity media
For using media as calibrations materials, low viscosity (LV) medium has been obtained by mixing regular cell growing medium (made by DMEM and 10% FBS) with 0.08% 6 kDa Dextran, while high viscosity (HV) medium has been obtained by mixing regular cell growing medium (made by DMEM and 10% FBS) with 6.48% 500 kDa Dextran[19].

For growing cells, we could not use these Dextran dispersed in regular growing medium, as cells died when cultured there for more than a few days. HV media for cell culture was made using respective complete growth media with the addition of 3.2% Polyvinylpyrrolidone (Sigma #P0930), which was biocompatible and ensured a final viscosity of 8 cP at 37 °C[19].

### Refractive index, density and shear viscosity measurements (Supplementary Table 2)
For every sample of Supplementary Table 2, repeating each measurement $n = 3$ times, refractive index $n$ has been obtained with a refractometer, density $\rho$ has been obtained by the ratio of 1 ml volume and its weight, while $\eta_{Shear}$ has been obtained by multiplying the dynamic viscosity (obtained from a Cannon–Fenske capillary viscometer) by the density; measurements of viscosity at 37 °C have been obtained by completely immersing the viscometer in a 37 °C water bath until temperature equilibrium.

N2 and S3 are Viscosity Reference Standards purchased from CANNON Instrument. Their refractive index, densities and viscosities come from Cannon Instrument datasheet.

### Live/dead analysis
Cells were either maintained in low viscosity (LV) media or switched to HV media 24 h after seeding and maintained in HV media for 7 days, passing when reaching ~70–80% confluency. On the 6th day, cells were seeded into 35 mm ibidiTreat (Ibidi) dishes at low density (20–30k cells). On the 7th day, one set of cells was incubated at room temperature for 1 h, and the live/dead viability assay (ThermoFisher) was performed immediately. The other set of cells was incubated at room temperature for 1 h and subsequently

returned to 37 °C for an additional 24 h of incubation to serve as a control for the cells in the first condition. After 24 h of incubation, the live/dead viability assay was performed. An additional group of cells was allowed to incubate at 37 °C for 48 h to serve as a control for the two previous conditions. After 48 h of incubation, the live/dead viability assay was performed. All fluorescence images were captured using a Zeiss 780 LSM confocal microscope with appropriate filters for calcein-AM (live dye) and ethidium homodimer-1 (dead cells). And homebuilt macro in ImageJ was used to count the number of live and dead cells and quantify morphological traits such as cell area, circularity, and aspect ratio.

### Brillouin microscopy imaging
A confocal Brillouin microscope was used for all experiments. Instrumentation details can be found in a previous report[36]. Briefly, a single-mode 660 nm continuous wave laser with an average power of ~50 mW was used. The laser beam was focused into the sample using a 40× objective lens (nominal NA = 0.95, Olympus) in a back-scattering configuration. The back focal aperture of the objective was overfilled by the beam with a beam expander[36]: in such a manner, the effective NA of our measurements was equal to the nominal NA.

The microscope spectral resolution was of 9 MHz for Brillouin shift and 18 MHz for Brillouin FWHMs, obtained by $N = 500$ Brillouin spectra of water (Supplementary Fig. 1). The backscattered Brillouin signal was collected using the same objective and analyzed using a two-stage cross-axis VIPA (Light Machinery, 15 GHz FSR) based spectrometer. The Brillouin spectra were recorded using an EMCCD camera (Andor, IXon Ultra 897) with an exposure time of 0.05 s.

Our imaging protocol matches other Brillouin imaging protocols available in the literature[33,36]. Briefly, to obtain 2D Brillouin maps of cells, YZ or XZ scans were acquired by scanning through the nucleus of the cell sample using a motorized stage (step size 0.5–1 μm). From there, the middle of the nucleus is identified as the optimal Z-level and XY Brillouin maps of the cells were acquired. Each acquisition used an exposure time of 0.05 s with a step size of 0.5 μm/pixel, for a total of ~2–3 min per acquisition. We stained cells for 20' with HOECHST and saw them live under transmission and fluorescence mode. Brightfield and fluorescence images were acquired immediately prior to Brillouin imaging; after each Brillouin acquisition, we checked the brightfield to ensure cells were stable, were not moving and did not show any thermal-induced damage like blebbing, which is a symptom of cell apoptosis[32,47]. Cells were replenished with fresh media right before imaging under the Brillouin Microscope in order to avoid osmolarity changes that might affect Brillouin shifts[49]. We fitted the spectra with a sum of 2 Lorentzians with a custom-made Matlab code. The nucleus of the cell in the Brillouin map was identified based on the fluorescence intensity. The cell in the Brillouin map was identified based on the Brillouin shift of the surrounding culture media. Extracted FWHM values were then corrected for instrumental broadening as in Eq. 3. The Brillouin shift and FWHM were averaged for the nuclear region of interest, identified with a binary mask obtained from the fluorescence intensity of the HOECHST signal.

### Stimulated Brillouin microscope imaging
The custom SBS spectrometer[41] used for these experiments consists of two CW single-frequency tunable lasers (a pump and a probe) at 780 nm (Toptica DL pro and TA pro), with ~100 kHz linewidth and 30–50 GHz tunability range, focused by two 40× objectives (nominal NA = 0.95) and overlapped in a counterpropagating geometry at the sample, which is a glass-bottom dish containing the reference liquid. The back focal aperture of the objective was overfilled by the beam with a beam expander; in such a manner, the effective NA of our measurements was equal to the nominal NA.

The pump frequency is locked at a Rb85 absorption line using a vapor-based locking module available with the laser, and its intensity is modulated at 1 MHz by an acousto-optic modulator (AOMO 3080-125, Crystal Technology, Inc.). SBS spectra are obtained by scanning the probe frequency in a 2–3 GHz window around the resonance with the acoustic wave, while recording the beat frequency between pump and probe with a fast detector

(1544-A, Newport) and a signal analyzer (N9000B, Keysight). The transmitted probe beam is detected by an amplified detector (PDA36A2, Thorlabs), from which the SBS signal is extracted using a bias tee (ZFBT-4R2GW+, MiniCircuits) and a lock-in amplifier (UHF, Zurich Instruments) referenced to the pump modulation frequency. A hot Rb85 vapor cell (Precision Glassblowing) in detection allows for the minimization of the background photons mainly coming from pump stray light, which, being modulated, is detected by the LIA.

Brillouin spectra were recorded by the data acquisition module present in the probe laser control unit, whose data sampling is synchronized with the probe frequency scanning. Spectra were acquired using around 100 mW pump power, around 20 mW probe power and 100 ms exposure time, to work in the shot-noise-limited regime of the instrument[41]. Hundred repetitions per sample were analyzed for statistics. In such a manner, the measurements lasted ~10 seconds, during which it is unlikely that we heated the sample.

SBM spectra shown in Fig. 1B, upper panel, are shown as converted to the laser wavelength of BM (660 nm) by multiplying the $x$-axis by a constant factor equal to $\frac{660}{780}$, i.e.: $v^{660nm} = v^{780nm} * \frac{660}{780}$. In such a manner, we obtained Brillouin shifts and widths of SBM as if we acquired them at 660 nm with the following equations: $v_B^{660nm} = v_B^{780nm} * \frac{660}{780}$; $\Gamma_B^{660nm} = \Gamma_B^{780nm} * \left(\frac{660}{780}\right)^2$.

## Statistics and reproducibility

For all the experiments of living cells, we always had at least $n = 3$ independent biological replicates. Statistical analysis was performed using GraphPad Prism. We always checked the normality of data with a Kolmogorov–Smirnov test. In the presence of multiple comparisons, ordinary one-way Anova using a Sidak correction for multiple comparisons was used, unless otherwise stated. A $p$-value of $p < 0.05$ was chosen as statistically significant. All the results shown in the graphs are given as mean ± standard deviation (SD).

## Reporting summary

Further information on research design is available in the Nature Portfolio Reporting Summary linked to this article.

## Data availability

Source data for graphs and charts that support the findings of this study are available in Numerical source data (Supplementary Data 1).

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

## Acknowledgements

This project has received funding from the European Innovation Council (EIC) through its Horizon Europe Pathfinder Program under grant agreement No. 101098989, ERC-2019-Synergy Grant (ASTRA, n. 855923), Project "National Center for Gene Therapy and Drugs based on RNA Technology" (CN00000041) financed by NextGeneration EU PNRR MUR-M4C2-Action 1.4-Call "Potenziamento strutture di ricerca e creazione di campioni nazionali di R&S" (CUP J33C22001130001) (to C.T.), and by the National Science Foundation (DBI1942003) and National Institutes of Health (R21CA258008, R01EY028666, R01EY030063). C.T. acknowledges financial support for this research by the Fulbright Research Scholar Program, which is sponsored by the U.S. Department of State and by the U.S.-Italy Fulbright Commission.

## Author contributions

Chenchen Handler: Data curation, Investigation, Visualization, Writing—original draft, Writing—review & editing. Giulia Zanini: Data curation. Ian M. Smith, Writing—review & editing. Kimberly M. Stroka: Writing—review & editing. Giuliano Scarcelli: Methodology; Conceptualization; Visualization, Writing—original draft; Writing—review & editing; Funding acquisition. Claudia Testi: Conceptualization, Data curation, Investigation, Visualization, Supervision, Writing—original draft, Writing—review & editing.

## Competing interests

The authors declare no competing interests.
