## [Transparent Peer Review file · Communications Biology]

Cellular mechanical properties in response to environmental viscosity imaged by Brillouin Microscopy

Corresponding Author: Dr Claudia Testi

Version 0:

Reviewer comments:

Reviewer #1

(Remarks to the Author)

The manuscript by Handler et al. reports a study on how cellular mechanical properties, specifically the longitudinal modulus and viscosity, respond to changes in extracellular fluid viscosity. The authors employed a custom-built confocal Brillouin Microscope (BM) to quantify the mechanical properties of non-cancerous breast cells (MCF10A) and highly metastatic breast cancer cells (MDA-MB-231) cultured in media with normal (0.77 cP) and high (8 cP) viscosities. To enable accurate viscosity measurements via the BM, the authors developed a new calibration method to deconvolve the instrumental broadening of the Brillouin spectra. By comparing ground truth Brillouin linewidths from calibration liquids with varying viscosities obtained using a stimulated Brillouin scattering microscope (SBM), the authors derived a reliable approach to correct for the linewidth broadening inherent in their confocal BM setup. This correction allowed for accurate extraction of the intrinsic linewidth, which serves as an indicator of the sample's longitudinal viscosity.

The biological study revealed that both cell types exhibited a decrease in nuclear longitudinal viscosity when cultured in high viscosity media, suggesting an adaptive response to their environment. Notably, while non-cancerous MCF10A cells maintained a consistent nuclear modulus regardless of extracellular viscosity, metastatic MDA-MB-231 cells showed an increase in nuclear modulus under high viscosity conditions. This finding implies that metastatic cells can modify their internal stiffness in response to changes in external fluid properties, potentially enhancing their ability to migrate and invade tissue. The study therefore proposes that variations in cellular viscosity and stiffness, as measured by deconvolved Brillouin widths, may serve as useful biomarkers for cancer metastasis.

This is a highly interesting and timely paper which provides novel methodological as well as biological findings. As such the results are interesting to the fields of Brillouin microscopy as well as (cellular) mechanobiology. Therefore I believe the study warrants publications, however I do have the following concerns and suggestions that I ask the authors to attend to before publication.

General points/concerns:

- Since the authors have an SBM that is not affected by linewidth broadening due to the spectrometer: Why did they not use the SBM for this study in the first place – i.e. use SBM to imaging the cells? This should be clarified and the approach clearly motivated and justified.
- The authors state that the role of viscosity in Brillouin Microscopy remains underexplored (Introduction, Discussion). However, there are several papers that have reported linewidth/viscosity measurements with BM that I believe should be cited and potentially discussed in this context. Eg. Margueritat et al. PRL 2019, Chan et al. Comms Bio 2021, and Beck et al. MBC 2024.
- I strongly encourage the authors to discuss the generalizability of the calibration method they propose. As far as I understand it is applicable to VIPA based spontaneous BM only. I think the authors should make this very clear, and also highlight that such calibrations depend on the BM implementation. Furthermore, the correct models for the fitting also highly depends on the optical design of the VIPA spectrometer.
- The interpretation and discussion of the biological aspects of the viscosity results requires more work. What are the true implications of less viscous cell nuclei for their behaviour, and potential disease properties? I assume such cells are less resistant to flow – how does this matter in a tumor/tissue/disease context?
- More generally, the authors should justify their focus on nuclear viscosity measurements. Why did they restricting their measurements and analysis to this sub-cellular structure? Did the authors attempt to measure the whole cell/cytoplasm?

Since the study is motivated by presumed changes in cell migration and spread, one would assume the cytoplasm (e.g. in the leading edge) to take an important role too.

- Table S1 shows that LV vs. HV also has different shift/longitudinal modulus. Can the authors confirm that the changes for the cancerous line is purely viscosity driven? Since the shift in Fig. 3B left seems also different between the cell lines.

Minor points:

- SBM and BM use different wavelengths (780 vs. 660nm) – this needs to be accounted for but I did not see this mentioned in the paper (maybe I missed it).

- I suggest to include a more comprehensive general introduction to the longitudinal modulus, and how it is different from commonly accepted moduli (e.g. Young's as measured by AFM, etc.). Also the frequency dependence and differences should be alluded to given the broad readership of the journal.

- The general reader would probably appreciate an explanation why Γ_{PSF} changes from day-to-day and with alignment.

- Fig. 1F shows a good correlation, but also a clear offset from the diagonal is visible – where does this come from?

- The authors note that high NA leads to line-broadening but that this in a backscattering configuration is known to be minimal. I would disagree as an NA of 0.95 is relatively high! Of course the effective NA depends on the fill factor – can the authors clarify this and discuss how the NA might affect both BM and SBM measurements?

- I recommend to clearly label in Fig.3 A the shift/width images with 'Brillouin shift/width'.

- Did the authors actually look into the fluorescence images in greater detail, i.e. the HOECHST images (not shown in Fig. 3A)? It might be interesting to see whether the DNA becomes more compacted and how this might spatially correlate with the shift/width maps.

- Line 35 Typo -> "intracellular"

- Line 171 The authors have shown that a Gaussian behaviour was the best fit, but here the Voigt model is included again.

- Line 369 I would not call this study "in vivo" but rather "in vitro".

- Line 115: The deconvolution operation does not affect the peak position only if f_{PSF} is symmetrical. This should be pointed out.

- Methods: Line 483 "We fitted the spectra with a sum of 2 Lorentzians with a custom-made Matlab code" Can the authors clarify whether the spectral fitting was done with Lorentzian or Gaussian (as I understood the latter gives more accurate linewidth results, as is the main point of this paper).

Reviewer #2

(Remarks to the Author)

This manuscript addresses a conceptually important topic, aiming to visualize for the first time the effects of viscosity on the mechanical properties of cells. While the reviewer acknowledges the potential significance and novelty of the work, the current presentation lacks sufficient rigor, technical justification, and clarity. The following points outline specific concerns that should be addressed to strengthen the manuscript:

1. Linewidth Measurement Methodology:

Measuring Brillouin linewidth using a conventional spectrometer is inherently challenging and generally less accurate than time-domain approaches, as extensively reviewed by Laubereau and Kaiser (<https://doi.org/10.1103/RevModPhys.50.607>). This limitation is also well recognized in the Brillouin microscopy community (e.g., [doi:10.1039/C5AN01700A](https://doi.org/10.1039/C5AN01700A), <https://doi.org/10.1364/OE.449980>, <https://doi.org/10.1364/BOE.10.001750>, <https://doi.org/10.1364/OE.487131>). Notably, the last two references focus specifically on improving the accuracy of Brillouin linewidth measurements. The current manuscript does not address these known challenges or provide sufficient justification for the chosen method, weakening the reliability of the reported viscoelastic measurements.

2. Use of Stimulated Brillouin Microscopy (SBM):

The rationale for employing a stimulated Brillouin microscopy system is not adequately justified. While SBM offers certain advantages, it does not inherently improve spectral resolution. Furthermore, the reported use of tightly focused 100 mW laser power is concerning. No thermal analysis is provided to estimate the temperature rise at the focal spot, which is critical, especially since previous studies (e.g., <https://doi.org/10.1073/pnas.2413938121>) have shown that even significantly lower intensities can be harmful at these excitation wavelengths. Given that viscoelastic properties are temperature-sensitive, it is unclear whether the measurements obtained under these conditions reflect intrinsic cellular properties or thermally induced artifacts.

3. Imaging Protocol and Temporal Resolution:

The imaging protocol lacks sufficient detail to assess the validity of the results. Living cells are inherently dynamic, and the observed spatial variations in viscoelastic properties shown in Figure 3 may reflect temporal fluctuations rather than spatial heterogeneity. The comparison with brightfield images is not convincing, as those are acquired much faster and represent integrated axial information, whereas Brillouin images are acquired over a longer duration and are restricted to a single optical section. More thorough explanation and controls are needed to validate the stability and reproducibility of the measured properties.

4. Data Interpretation and Analysis:

The presence of prominent "hot spots" in the Brillouin images raises questions about data consistency and interpretation. The manuscript does not address whether these features are artifacts, localized structural elements, or influenced by imaging parameters (e.g., focal plane, axial position). A discussion of their origin and potential impact on data interpretation is necessary to support the reliability of the conclusions.

Overall, while the study has potential to contribute meaningfully to the field, the current manuscript requires major revisions to clarify experimental design, validate measurement techniques, and improve the rigor of data interpretation.

Version 1:

Reviewer comments:

Reviewer #1

(Remarks to the Author)

I have read the revised version of the paper as well as the rebuttal by the authors. I appreciate the detailed point-by-point response which thoroughly addressed and clarified all my questions and minor concerns. The paper has increased in clarity and I recommend publication as is.

Robert Prevedel, EMBL Heidelberg

I have carefully read and assessed the responses of the author to the other Rev. #2, who, overall, found the study to potentially contribute to the field, but suggested major revisions to clarify experimental design, validate measurement techniques, and improve the rigor of data interpretation. Below is my unbiased assessment of how the authors have addressed these points:

- 1) Linewidth Measurement Methodology: Here the authors now discuss possible limitations of their work and how their approach helps to overcome them. They now also provide references to the article suggested by the Reviewer. In my view this point is fully addressed.
- 2) Use of SBM: The authors clarified that SBM and the associated high pump powers were not used for biological imaging, and only for calibration. Therefore the suggested thermal analysis is not needed in my view.
- 3) Imaging Protocol: The Reviewer was concerned about potential cell movement and that BM is too slow in relation. The authors clarified that their imaging protocol also involved confocal fluorescence and bright field imaging before/after the BM, and that cells did not move during acquisition.
- 4) Data Interpretation and Analysis: The reviewer was concerned about hotspots present in the BM images. The authors elaborated that these are sub-cellular structures that can also be seen in brightfield and other published BM images, and thus most likely not artefacts. Based on personal experience I fully agree with this assessment. Furthermore, these are cytoplasm structures and thus don't influence the results/analysis that was focused on the cell nucleus.

Open Access This Peer Review File is licensed under a Creative Commons Attribution 4.0 International License, which permits use, sharing, adaptation, distribution and reproduction in any medium or format, as long as you give appropriate credit to the original author(s) and the source, provide a link to the Creative Commons license, and indicate if changes were

made.

Response to Reviewers

We thank the reviewers for their careful evaluation of our manuscript and their thoughtful, constructive feedback. Their comments have been invaluable in enhancing the quality and impact of our work. In response, we have made a concerted effort to address each of their questions and to provide comprehensive clarifications where needed.

In the revised manuscript all changes have been highlighted in **yellow**. We have incorporated all the reviewers' comments and suggestions. In line with the journal's editorial policies, we updated the figures displaying mean values by either enlarging or adding individual data points to better illustrate the data distribution. Furthermore, we now provide full access to the underlying numerical data for all figures and charts through a newly added Supplementary Data File S5, which contains the complete raw dataset in Excel format.

Please find attached a detailed, point-by-point response to all reviewer comments. Reviewer remarks are presented in **bold**, with our responses in *italics*. All line references correspond to the revised version of the manuscript.

Reviewer #1:

The manuscript by Handler et al. reports a study on how cellular mechanical properties, specifically the longitudinal modulus and viscosity, respond to changes in extracellular fluid viscosity. The authors employed a custom-built confocal Brillouin Microscope (BM) to quantify the mechanical properties of non-cancerous breast cells (MCF10A) and highly metastatic breast cancer cells (MDA-MB-231) cultured in media with normal (0.77 cP) and high (8 cP) viscosities. To enable accurate viscosity measurements via the BM, the authors developed a new calibration method to deconvolve the instrumental broadening of the Brillouin spectra. By comparing ground truth Brillouin linewidths from calibration liquids with varying viscosities obtained using a stimulated Brillouin scattering microscope (SBM), the authors derived a reliable approach to correct for the linewidth broadening inherent in their confocal BM setup. This correction allowed for accurate extraction of the intrinsic linewidth, which serves as an indicator of the sample's longitudinal viscosity.

The biological study revealed that both cell types exhibited a decrease in nuclear longitudinal viscosity when cultured in high viscosity media, suggesting an adaptive response to their environment. Notably, while non-cancerous MCF10A cells maintained a consistent nuclear modulus regardless of extracellular viscosity, metastatic MDA-MB-231 cells showed an increase in nuclear modulus under high viscosity conditions. This finding implies that metastatic cells can modify their internal stiffness in response to changes in external fluid properties, potentially enhancing their ability to migrate and invade tissue. The study therefore proposes that variations in cellular viscosity and stiffness, as measured by deconvolved Brillouin widths, may serve as useful biomarkers for cancer metastasis.

This is a highly interesting and timely paper which provides novel methodological as well as biological findings. As such the results are interesting to the fields of Brillouin microscopy as well as (cellular) mechanobiology. Therefore I believe the study warrants publications, however I do have the following concerns and suggestions that I ask the authors to attend to before publication.

We thank the reviewer for recognizing the significance and thoughtful design of our study. We hope that the following responses address all the concerns raised and contribute to enhancing the quality of the manuscript, which has significantly improved thanks to the insightful comments and suggestions from both reviewers.

General points/concerns:

- Since the authors have an SBM that is not affected by linewidth broadening due to the spectrometer: Why did they not use the SBM for this study in the first place – i.e. use SBM to imaging the cells? This should be clarified and the approach clearly motivated and justified.

We thank the reviewer for this comment. While SBM is unaffected by linewidth broadening and certainly has some advantages compared to spontaneous Brillouin microscopy, such as the absence of the elastic scattering signal and high spectral resolution in certain cases, spontaneous Brillouin microscopy is currently the standard for label-free mechanical measurements for biology applications. Additionally, spontaneous Brillouin microscopy is more widely available and inexpensive compared to Stimulated Brillouin, that requires 2 laser sources (pump and probe) and often tunable laser sources. SBM is also far more sensible to the optical alignment of the counter-propagating lasers than the Spontaneous, rendering it more complicated to use on highly scattering materials such as cells. Moreover (and even more importantly as the second Reviewer rightly pointed out), the use of SBM at such high powers on living cells might induce serious thermal damages (Li et al., PNAS 2024, doi: 10.1073/pnas.2413938121; Shaashoua et al., Nat. Photonics 2024, doi: 10.1038/s41566-024-01445-8; Bilenca et al., JPHys Photonics 2024, doi: 10.1088/2515-7647/ad5506). A possible way to minimize photodamage would be with a Pulsed Stimulated Brillouin Microscope (Yang et al., Nat Methods 2023, doi: 10.1038/s41592-023-02054-z) by using a pulsed pump laser, but this was not accessible in our lab.

It is well known that the use of a spectrometer as in spontaneous BM introduces spectral broadening and thus, the focus of our work was to develop a straightforward and relatively simple method of utilizing intrinsic widths of 4 calibration materials (obtained with SBM) to deconvolve the Brillouin spectra acquired with spontaneous Brillouin Microscopes. In such a manner, the use of SBM was intended just to provide a “database” of intrinsic widths to be used on BM data, not to use it on living cells.

We have added the following to the Discussion to expand on this point (lines 299-302 of the revised manuscript): “In light of this, we sought to develop a generalizable and straightforward method to deconvolve the Brillouin spectra acquired with spontaneous Brillouin microscopes by requiring only the addition of two extra standards to the calibration protocol established in literature^{33,36}.”

Then, we now continue with a clear motivation and justification of our approach (lines 306-321): “In such a manner, we used linewidth data obtained from SBM solely to deconvolve the spectra acquired with our BM on cells. Importantly, SBM was not applied to biological samples, as its use in this context remains limited, due to the high laser powers involved,

which can cause thermal damage to cells^{32,55,56}. In contrast, spontaneous Brillouin microscopy is currently the standard technique for probing cellular and tissue biomechanics⁵⁷; it is more widely adopted, less expensive, and better suited for biological applications than SBM.”

- The authors state that the role of viscosity in Brillouin Microscopy remains underexplored (Introduction, Discussion). However, there are several papers that have reported linewidth/viscosity measurements with BM that I believe should be cited and potentially discussed in this context. Eg. Margueritat et al. PRL 2019, Chan et al. Comms Bio 2021, and Beck et al. MBC 2024.

We sincerely thank the reviewer for this insightful comment and for suggesting valuable references reporting linewidth measurements. Our intention was to highlight that, compared to the Brillouin shift, linewidth and viscosity measurements remain relatively underexplored and underreported, particularly in studies using Brillouin spectrometers to investigate the mechanical properties of cancer cells.

In this context, the study by Margueritat et al. is particularly relevant, as it reports both the Brillouin shift and linewidth of colorectal carcinoma spheroids derived from SW480 and HCT116 cell lines. Interestingly, the authors observed no significant difference in Brillouin shift between the two cell types—consistent with similar refractive indices and storage moduli—while they reported statistically significant differences in linewidth, with HCT116 spheroids exhibiting lower linewidths than SW480. The authors attribute this difference to the distinct metastatic potential of the two lines, with SW480 representing early-stage colon cancer and HCT116 associated with more advanced, metastatic. Notably, these findings are consistent with our own results, supporting the idea that increased tumor aggressiveness may correlate with lower Brillouin linewidths—and, by extension, lower longitudinal viscosities. While Margueritat et al. used a low-NA objective (0.35), thereby minimizing objective-induced spectral broadening, our results show the same trend despite using a higher-NA objective. This consistency suggests that our deconvolution process, calibrated using SBM data acquired under matched NA conditions, effectively corrects for such broadening. However, we note that the linewidth values reported by Margueritat et al. were extracted from raw Brillouin spectra without applying any deconvolution. As recently emphasized in the “Consensus paper for Brillouin microscopy” (Bouvet et al., Nature Photonics, 2025, doi: 10.1038/s41566-025-01681-6), such an approach tends to significantly overestimate linewidths (for example, leading to inflated viscosity estimates for water by several-fold). Although their use of a tandem Fabry–Pérot interferometer provides higher spectral resolution than our double-VIPA setup, some degree of instrumental broadening remains inevitable and should still be corrected through deconvolution. Finally,

our data reveal notable day-to-day variability in the BM vs. SBM calibration curves, likely due to changes in spectrometer alignment. This further underscores the importance of performing daily deconvolution to eliminate alignment-related noise and improve the reliability of linewidth and viscosity measurements. We now reference these important findings and their alignment with our results in lines 460-465 of the revised manuscript: “These findings are also consistent with other studies⁵⁰ reporting lower Brillouin linewidths in cells from more aggressive metastatic stages compared to early-stage cancer cells. This supports the potential use of Brillouin linewidth as a parameter for distinguishing tumor tissues based on metastatic potential. Notably, that study employed an objective with a significantly lower numerical aperture objective than ours, suggesting that the broadening induced in our data by the high NA does not compromise the validity of our deconvolution protocol.”

Chan et al. (Comms Bio 2021) analyzed the Brillouin shift and Brillouin loss tangent (related to linewidth) during follicle development, identifying differences across developmental stages. The context is different from ours, so a comparison of the results might be difficult; also, the authors deeply discuss the Brillouin loss tangent more than merely the Brillouin shift or width. From a technical point of view, their approach assumes that the spectrometer response follows a Lorentzian profile, and they perform deconvolution by simply subtracting the instrument linewidth from the fitted linewidth. Unlike our method, this does not involve validation against intrinsic linewidths, which may lead to underestimation of the true values. Additionally, their deconvolution uses a fixed spectrometer linewidth, whereas our findings demonstrate that this value can vary from day to day depending on spectrometer alignment. We already addressed this issue in our manuscript (lines 350–354), emphasizing the need for daily calibration to ensure accurate deconvolution: “Other works in literature show a deconvolution of Brillouin spectra by subtracting a fixed Γ_{PSF} from Brillouin widths^{59,61}, assuming a Lorentzian relationship between $\Gamma_{experimental}$ and Γ_B for their spectrometer, but our results show that this procedure might result in a severe underestimation of the retrieved Γ_B if the comparison with theoretical values is not performed”.

Finally, Beck et al. (Molecular Biology of the Cell, 2024) reported differences in Brillouin shifts and linewidths in *in vitro*-generated protein condensates. However, similar to Margueritat et al., the authors did not perform spectral deconvolution. In this case, the issue may be even more pronounced, as their use of a double-VIPA setup, having broader instrumental responses than a tandem Fabry–Pérot, likely results in a substantial overestimation of linewidths. As previously noted in the Brillouin microscopy consensus paper (Bouvet et al., Nature Photonics, 2025, doi: 10.1038/s41566-025-01681-6), neglecting deconvolution can lead to significant errors, including overestimating values by several fold.

In the revised version of the manuscript, we have cited the recommended manuscripts and now discussed their findings. We have added the following to the discussion section (lines 290-295): “The role of viscosity in Brillouin Microscopy remains relatively underexplored in the current literature, where many only report the Brillouin frequency shift corresponding to the elastic or “storage” modulus. While some do co-report both Brillouin frequency shift and Brillouin linewidth^{50,51}, they do not perform spectral deconvolution, which is a necessary step to accurately isolate the intrinsic linewidth associated with viscous properties.”

We also added the following (lines 354-359): “Additionally, using a fixed spectrometer linewidth for deconvolution overlooks day-to-day variability due to changes in system alignment, a factor that should be accounted for. Other studies, furthermore, present valuable insights into Brillouin linewidths, but do not perform any deconvolution on the spectra^{50,51}: as recently highlighted³³, this leads to systematic overestimation of linewidths, failing to correct for instrumental broadening and thermal drifts that occur between measurement sessions.”

- I strongly encourage the authors to discuss the generalizability of the calibration method they propose. As far as I understand it is applicable to VIPA based spontaneous BM only. I think the authors should make this very clear, and also highlight that such calibrations depend on the BM implementation. Furthermore, the correct models for the fitting also highly depends on the optical design of the VIPA spectrometer.

We thank the reviewer for pointing out this important aspect, that was missing in our first version. The deconvolution approach presented in this manuscript was developed for a state-of-the-art spontaneous Brillouin Microscope equipped with a double-VIPA spectrometer. However, in principle this method can be extended to other spectrometer designs as well: users can apply the same strategy by plotting the FWHM obtained from their own spectrometer against our SBM linewidth values (which are free from dispersion-related broadening and thus represent the true linewidths of the calibration materials, shown in Table S1). If a different laser wavelength is used, our SBM linewidth values can be appropriately scaled, as described in the Methods section, making the deconvolution approach adaptable to Brillouin microscopes operating at different wavelengths. Using the four calibration materials, other researchers can apply various fitting models (e.g., linear, root mean square, Voigt) to determine the most suitable relationship between their spectrometer response and the intrinsic linewidths. These models arise from convolution mathematics and are independent of the specific spectrometer design: in fact, the original reference (Hermans et al., OPTICA, 2020, which is the reference #37 in the revised

manuscript) applied this concept in the context of Terahertz spectroscopy, further illustrating its generality.

The only specific dependency in our calibration protocol is the NA of the objective used to focalize the laser beam on the sample, which was 0.95 for both BM and SBM measurements. To maintain consistency with our calibration dataset, users should acquire spectra of the calibration materials using an objective with the same NA. Using a different NA can introduce spectral broadening and even slight shifts in Brillouin peak position (Antonacci et al., APL, 2013, doi: 10.1063/1.4836477).

We have added the following to the discussion section (lines 390-398): “Notably, this deconvolution approach is designed to be broadly applicable to different Brillouin Microscopes. By measuring the Brillouin spectra of the four calibration materials, users can determine the relationship between their experimental linewidths and our intrinsic values obtained with the SBM (Table S1), enabling spectral deconvolution specific to their system. Importantly, this method is not dependent on the specific optical design of the spectrometer nor on the wavelength of the laser source. The only critical parameter is the NA of the objective, which should match the one used in our study (NA = 0.95): a different NA may introduce spectral broadening and shifting, potentially affecting the reliability of the deconvolution⁶².”

- The interpretation and discussion of the biological aspects of the viscosity results requires more work. What are the true implications of less viscous cell nuclei for their behaviour, and potential disease properties? I assume such cells are less resistant to flow – how does this matter in a tumor/tissue/disease context?

We thank the reviewer for this question. Ultimately, we aren't entirely sure at this time, as this field is a relatively new area of study that is being increasingly explored. However, we can start to make interpretations regarding what these implications mean.

Firstly, we need to reconsider what viscosity means on a molecular level, not simply as a resistance to flow, but a resistance to movement as a whole. Cells are primarily composed of water, and their cytosolic or nuclear makeup causes that water to behave structurally differently, resulting in a more gel-like consistency, as opposed to a traditional liquid (Hertzog et al., Cells 2023; doi: 10.3390/cells12151958). So, as we have an increase in the viscosity of the nuclei, we can assume that there is a resulting increase in the resistance to their motility. There are numerous ways of cell motility, but they all depend on the intracellular movement of cytosolic components (Paul et al., Nat. Rev. Cancer 2017, doi: 10.1038/nrc.2016.123); thus, if a cell's contents are less viscous, they will require less energy

to move during phenomena such as cell motility. Moreover, a decrease in nuclear viscosity suggests that the cell nucleus becomes more deformable and less resistant to mechanical stresses. However, when focusing on metastatic cells, this is particularly relevant in the context of tumor progression: cancer cells are required to migrate through dense and mechanically confining microenvironments, such as the extracellular matrix and endothelial barriers during intravasation and metastasis. The nucleus is typically the largest and stiffest organelle in the cell, and its mechanical properties can be rate-limiting for migration through confined spaces. As such, reduced nuclear viscosity may facilitate nuclear deformation and enhance a cell's ability to squeeze through tight interstitial pores and vessel walls, thereby promoting invasive behavior and metastatic dissemination. When we investigated the mechanical properties of these mammary cells, which have an increased metastatic potential (i.e., more aggressive invasion and movement potential), they adopted a lower viscosity in their nucleus, likely to facilitate movement.

We have added the following to the discussion section to expand on this aspect, while adding more references to support our statements (lines 436-451): “This result is interesting, as cellular motility is dependent on intracellular movements of cytosolic components and subsequently nuclear components⁷³; thus, if a cell's contents are less viscous, they will require less energy to migrate⁷⁴. Moreover, a decrease in nuclear viscosity suggests that the cell nucleus may become more deformable and less resistant to mechanical stresses. This is particularly relevant in the context of tumor progression: our results thus might provide an important biomechanical motivation to the enhanced motility seen in cancer cells in highly viscous extracellular fluids^{21,22,72}. Indeed, metastatic cells migrate through crowded and confined microenvironments, such as the extracellular matrix and endothelial barriers during intravasation and extravasation^{1,2,7,15,73}. The nucleus is the largest and stiffest organelle in the cell, and its mechanical properties are the limiting factor for the deformability and migration of the whole cell through confined spaces^{1,2,7,15,73}: reduced nuclear viscosity may thus facilitate nuclear deformation and enhance a cell's ability to squeeze through tight pores and vessel walls, thereby promoting invasive behavior and metastatic dissemination. Furthermore, decreased nuclear viscosity influences gene expression by activating different mechanotransduction pathways, as already shown⁷².”

We also added the following phrase (lines 472-474): “In this context, it will be of interesting future work to deepen why cancer cells retain low viscosity while adapting their internal stiffness to the environment, consistent with previous observations⁴⁷.”

- More generally, the authors should justify their focus on nuclear viscosity measurements. Why did they restricting their measurements and analysis to this sub-

cellular structure? Did the authors attempt to measure the whole cell/cytoplasm? Since the study is motivated by presumed changes in cell migration and spread, one would assume the cytoplasm (e.g. in the leading edge) to take an important role too.

We thank the reviewer for this comment. Our protocol for cellular Brillouin acquisitions is refined to give us the best representation of the nucleus of the cells. This is because the nucleus is the largest organelle in the cell and therefore interesting from a mechanical standpoint as it is a limiting factor in the deformability of cells. We first perform a XZ or YZ scan to determine the middle of the nucleus of the cell then take an XY image at the plane that best represents the middle of the nucleus. Though we are still able to capture the cytoplasm in this plane, it doesn't necessarily represent the cytoplasm at the best.

To capture the cytoplasm better, we would have to image much closer to the growing substrate; however, Brillouin Microscopy has its own difficulties in imaging close to the cell-substrate interface, that can introduce a lot of spurious noise due to unfiltered Rayleigh signal, seen in the resulting Brillouin images as in the following figures:

In the revised version of the manuscript, we provide a significant motivation on why we focused on the nucleus mechanics (lines 440-449): “ This is particularly relevant in the context of tumor progression: our results on nuclear viscosity thus might provide an important biomechanical motivation to the enhanced motility seen in cancer cells in highly viscous extracellular fluids^{21,22,72}. Indeed, metastatic cells migrate through crowded and confined microenvironments, such as the extracellular matrix and endothelial barriers during intravasation and extravasation^{1,2,7,15,73}. The nucleus is the largest and stiffest organelle in the cell, and its mechanical properties are the limiting factor for the deformability and migration of the whole cell through confined spaces^{1,2,7,15,73}: reduced nuclear viscosity may

thus facilitate nuclear deformation and enhance a cell's ability to squeeze through tight pores and vessel walls, thereby promoting invasive behavior and metastatic dissemination”.

Anyway, we understand that the restriction to the nucleus may represent a limitation of our study and we have added the following lines to the revised version of the manuscript (lines 486-494): “A potential limitation of this study is that our analysis was focused on the nucleus, which is the largest and stiffness organelle in the cell: it is therefore interesting from a mechanical standpoint, representing the limiting factor in cellular deformability. Moreover, our protocol for cellular Brillouin acquisitions is refined to give us the best representation of the nucleus of the cells. Extending this analysis to the cytoplasm could offer additional insights, especially considering its role in cell motility and spread. However, our Brillouin measurements in the cytoplasm were limited by high noise: this was due to its proximity to the growing substrate, where reflections from the glass surface produced strong Rayleigh scattering, interfering with spectral acquisition.”

- Table S1 shows that LV vs. HV also has different shift/longitudinal modulus. Can the authors confirm that the changes for the cancerous line is purely viscosity driven? Since the shift in Fig. 3B left seems also different between the cell lines.

We thank the reviewer for this comment. This is an interesting point and something we noticed, too. We were careful that no other parameters were changed as all cells were seeded on the same type of dishes (35 mm ibidiTreat dishes) and we made sure that the osmolarity of the low viscosity and high viscosity medias were the same (Figure 2C). The only thing that differed was the viscosity of the media and therefore suggests that any changes to cellular elastic and viscous modulus are purely driven by the heightened extracellular fluid viscosity.

However, this is a topic that is still be investigated but others have reported that heightened extracellular viscosity increases cell migration speeds, and our collaborators have seen that cells have higher deformability when cultured in high viscosity media (manuscript under review at FEBS).

Minor points:

- SBM and BM use different wavelengths (780 vs. 660nm) – this needs to be accounted for but I did not see this mentioned in the paper (maybe I missed it).

We apologize if this important detail was not provided in the text.

In the revised manuscript, we provided this information:

- *in the main text (lines 160-162): “The laser source of SBM operated at 780 nm, while BM worked at 660 nm; to ensure consistency, all results were converted to the reference wavelength of BM with a scaling factor (Methods)”, lines 171 and 180: “We show their Brillouin widths (Figure 1D left) and shifts (Figure 1D right), both converted to the wavelength of the BM”.*
 - *in the caption of Figure 1B and 1C (lines 856-857): “SBM laser source operated at 780 nm; to facilitate comparison with BM results, the spectra are here shown to an equivalent wavelength of 660 nm (described in the Methods)”. And lines 859-860: “BM laser source was 660 nm: we used this wavelength as a reference for all the measurements.”*
 - *In the Methods section (lines 619-622): “SBM spectra shown in Figure 1B, upper panel, are shown as converted to the laser wavelength of BM (660 nm) by multiplying the x axis for a constant factor equal to 660/780: i.e., $v^{(660\text{ nm})} = v^{(780\text{ nm})} * 660/780$. In such a manner, we obtained Brillouin shifts and widths of SBM as if we acquired them at 660 nm with the following equations: $v_B^{(660\text{ nm})} = v_B^{(780\text{ nm})} * 660/780$; $\Gamma_B^{(660\text{ nm})} = \Gamma_B^{(780\text{ nm})} * (660/780)^2$ ”*
- I suggest to include a more comprehensive general introduction to the longitudinal modulus, and how it is different from commonly accepted moduli (e.g. Young’s as measured by AFM, etc.). Also the frequency dependence and differences should be alluded to given the broad readership of the journal.**

We thank the reviewer for pointing out this important aspect, and we fully agree that providing an overview of the different mechanical moduli would enhance the clarity of the manuscript. In response, we have also included a comparison of the high-frequency nature of Brillouin scattering with other conventional mechanical testing techniques. We have added the following (lines 100-108 of the revised version):

“The Longitudinal Modulus represents the ratio of uniaxial stress to strain whereby the material experiences a change in volume, differently than the more commonly used Young’s modulus, where the material volume is kept constant. Moreover, because Brillouin probes material properties at high frequencies (GHz regime), the measured modulus is reported in the GPa range; other mechanical testing methods (i.e. Atomic Force Microscopy, AFM) use much lower probing frequencies and the measured modulus is in the Pa – kPa range²⁶. While the longitudinal modulus M and Young’s modulus E are fundamentally different, they both follow the same direction of deformation and empirical relationships between M and E have been established for specific experimental conditions^{25,34}.”

- The general reader would probably appreciate an explanation why Γ_{PSF} changes from day-to-day and with alignment.

We thank the reviewer for highlighting this aspect that was missing in our previous version. Lasers used in Brillouin Microscopy have the tendency to drift over time due to room temperature fluctuations. Moreover, also the VIPAs in the spectrometer are subjected to room temperature which can lead to day-to-day changes in system alignment (Bouvet et al., Nature Photonics 2025, doi: 10.1038/s41566-025-01681-6; Testi et al., arXiv:2412.20516, doi: 10.48550/arXiv.2412.20516). These variations are typically reflected in the Brillouin linewidth maps, where the minimum and maximum values may fluctuate across different days, while the Brillouin shift maps remain consistent.

To improve clarity, we have added the following statement to the Results section (lines 149-153 of the revised manuscript): “Moreover, Γ_{PSF} can vary from day to day due to factors such as room temperature fluctuations, laser and VIPAs drifting^{33,40}, and differences in spectrometer alignment between users or across different sessions. Since alignment often requires repositioning the VIPAs, a deconvolution approach should be used that does not rely on a fixed value of Γ_{PSF} .”

We also added the following sentence to the Results (lines 201-202): “... due to room temperature fluctuations, confirming the need of a non-constant value of Γ_{PSF} ”.

- Fig. 1F shows a good correlation, but also a clear offset from the diagonal is visible – where does this come from?

The referee is right, a clear offset is visible, we apologize for not mentioning it in the original version of the manuscript. We are not sure where this offset comes from, because the retrieved η_{Bulk} depends on multiple parameters that we experimentally and independently obtained: namely, η_{shear} with a viscometer, n with a refractometer and the deconvolved Γ_B with our method. Among these, we believe the most likely source of the offset might be a systematic overestimation of Γ_B as it is extrapolated from fits that inherently carry uncertainty, especially at lower linewidth values. However, potential measurement errors in η_{shear} and n , or small inconsistencies in the experimental conditions could also contribute.

In any case, the deviation from literature values is within or slightly below the error bar, suggesting it is not statistically significant. Furthermore, the literature values used for comparison were also experimentally derived, but error bars were not reported: this lack of uncertainty quantification in the reference data may also influence the apparent offset on the x axis.

We have now acknowledged and discussed this point in the revised manuscript (lines 381-385): “The difference between our experimentally retrieved η_{bulk} values was within two standard deviations from values found in literature. Here, an offset from the diagonal was observed, likely arising from a slight overestimation of Γ_B , but the other terms such as n , η_{shear} – experimentally retrieved with different techniques - might also contribute; additionally, the absence of error bars in the literature values could influence the observed deviation on the x axis.”

- The authors note that high NA leads to line-broadening but that this in a backscattering configuration is known to be minimal. I would disagree as an NA of 0.95 is relatively high! Of course the effective NA depends on the fill factor – can the authors clarify this and discuss how the NA might affect both BM and SBM measurements?

We thank the reviewer for this comment and we apologize if our original explanation was unclear. While we agree that an NA of 0.95 is very high, the broadening we determined experimentally was relatively small (between 530-730 MHz). In Brillouin Microscopy not only the NA of the objective, but also the scattering angle has an incidence on the broadening of the peaks: what we intended to note is that, although high NA contributes to Brillouin spectral broadening and shift, the choice of a backscattering (180°) geometry in our setup minimizes this effect compared to other collection angles. Therefore, while our NA is indeed high, using a backscattering configuration in both BM and SBM mitigates additional angular broadening contributions: in a configuration that is not 180° the broadening would thus be even higher.

Importantly, both our BM and SBM systems use the same objective (NA = 0.95) and operate in a backscattering configuration, ensuring that the influence of NA is consistent across both modalities. This consistency allows us to isolate the contribution of the spectrometer to spectral broadening (which is present only in BM) as the primary target for deconvolution. Thus, while the high NA does contribute to spectral broadening, this effect was identical in both BM and SBM. The additional broadening observed in BM arises from the spectrometer, which was the contribution we sought to remove from our data. The good agreement between our deconvolved data and literature viscosity values (Figure 1F) supports the validity of this approach.

Regarding the fill factor, we ensured that the back aperture of the objective was fully filled using a beam expander, as described in our previous protocol (Zhang et al., Nat. Protoc., 2021, doi: 10.1038/s41596-020-00457-2). This guarantees that the effective NA matched the nominal NA of 0.95 during all measurements.

We better discussed these aspects in the revised manuscript (lines 368-371) by adding several details: “It is important to note that the use of a high numerical aperture objective also broadens and shifts the Brillouin peak in both BM and SBM; however, in a backscattering configuration like ours, this effect is known to be minimal with respect to other scattering angles, that would induce a higher broadening⁶². To avoid this influence, we acquired Brillouin spectra in both BM and SBM using the same objective and in the same backscattering configuration, thereby correcting BM widths only for the spectrometer presence.”

In the methods, we better described the filling factor of both BM and SBM setups (lines 569-572 and 599-602): “The laser beam was focused into the sample using a 40x objective lens (nominal NA = 0.95, Olympus) in a back-scattering configuration. The back focal aperture of the objective was overfilled by the beam with a beam expander³⁶: in such a manner, the effective NA of our measurements was equal to the nominal NA.”

- I recommend to clearly label in Fig.3 A the shift/width images with ‘Brillouin shift/width’.

We thank the reviewer for this comment and have added the appropriate labels to Figure 3A, which is here represented:

- Did the authors actually look into the fluorescence images in greater detail, i.e. the HOECHST images (not shown in Fig. 3A)? It might be interesting to see whether the DNA

becomes more compacted and how this might spatially correlate with the shift/width maps.

We thank the reviewer for raising this interesting comment. We only used HOECHST to stain the nucleus as a guide for analyzing the nuclear Brillouin shift and FWHM. To investigate how compact the DNA is, we would have to do much more in depth nuclear staining which was beyond the scope for this manuscript but would definitely be an interesting direction to explore in the future.

- Line 35 Typo -> "intracellular"

We thank the reviewer for this comment and have made the appropriate correction to the typo.

- Line 171 The authors have shown that a Gaussian behavior was the best fit, but here the Voigt model is included again.

We thank the reviewer for this comment and realized this was a typo. We have made the appropriate correction to this sentence, which now is (Lines 193-195):

"These results confirmed that the behavior between $\Gamma_{\text{experimental}}$ vs Γ_B was not linear nor Voigt and that the VIPAs were mostly described by a Gaussian."

- Line 369 I would not call this study "in vivo" but rather "in vitro".

*We thank the reviewer for this comment and have made the appropriate corrections to the typo (now line 431): "Our in vitro Brillouin Microscopy data (**Figure 3**) show that..."*

- Line 115: The deconvolution operation does not affect the peak position only if f_{PSF} is symmetrical. This should be pointed out.

We thank the reviewer for this detail, which we mistakenly did not mention. We corrected it in the revised version (new line 132): "If $f_{\text{PSF}}(x)$ is symmetrical, the convolution operation affects only the widths, not the peaks position".

- Methods: Line 483 "We fitted the spectra with a sum of 2 Lorentzians with a custom-made Matlab code" Can the authors clarify whether the spectral fitting was done with Lorentzian or Gaussian (as I understood the latter gives more accurate linewidth results, as is the main point of this paper). We thank the reviewer for this comment. To clarify, the experimental spectra were fitted using

a sum of 2 Lorentzian functions, consistent with the theoretical expectation that Brillouin peaks follow a Lorentzian lineshape. These fits were performed using a custom MATLAB script. For completeness, we also tested Gaussian fits on representative spectra and found that the resulting parameters were nearly identical; however, all reported values are based on Lorentzian fits, as this is the current standard in the literature. In contrast, the elastic peaks were best fitted using a sum of two Gaussians rather than Lorentzians, as demonstrated in Supplementary Figure S2. This likely reflects the sharper behavior of the elastic peak, which is better represented by the steeper tails of a Gaussian function, while Lorentzian functions are more suitable for the broader sample Brillouin peaks.

Reviewer #2:

This manuscript addresses a conceptually important topic, aiming to visualize for the first time the effects of viscosity on the mechanical properties of cells. While the reviewer acknowledges the potential significance and novelty of the work, the current presentation lacks sufficient rigor, technical justification, and clarity.

We thank the reviewer for this comment and we apologize for the insufficient justification of the method. We are sure that the revised version of the manuscript has been improved in clarification and rigor.

The following points outline specific concerns that should be addressed to strengthen the manuscript:

1. Linewidth Measurement Methodology:

Measuring Brillouin linewidth using a conventional spectrometer is inherently challenging and generally less accurate than time-domain approaches, as extensively reviewed by Laubereau and Kaiser (<https://doi.org/10.1103/RevModPhys.50.607>). This limitation is also well recognized in the Brillouin microscopy community (e.g., doi: [10.1039/C5AN01700A](https://doi.org/10.1039/C5AN01700A), <https://doi.org/10.1364/OE.449980>, <https://doi.org/10.1364/BOE.10.001750>, <https://doi.org/10.1364/OE.487131>). Notably, the last two references focus specifically on improving the accuracy of Brillouin linewidth measurements. The current manuscript does not address these known challenges or provide sufficient justification for the chosen method, weakening the reliability of the reported viscoelastic measurements.

We thank the reviewer for this comment and we apologize if the original manuscript did not provide sufficient justification for the methodology we employed. In the revised version of the manuscript, we have included citations to the recommended literature regarding the use of time-domain approaches as efficient methods for measuring Brillouin linewidth, and we have recognized these challenges as a possible limitation of our work.

That said, the application of time-domain techniques to living biological samples remains relatively uncommon in most laboratory settings, which is why we opted for spontaneous Brillouin microscopy operating in the frequency domain. We would also like to emphasize that, when appropriate deconvolution is applied, reliable linewidths can be extracted from frequency-domain Brillouin spectra. This is clearly illustrated in Figure 3B of the recently published “Consensus statement on Brillouin light scattering microscopy in biological

materials” (Bouvet et al., *Nature Photonics*, 2025, doi: 10.1038/s41566-025-01681-6), which became available after our initial submission.

We have added to following phrases to the introduction (lines 74-80 of the revised version of the manuscript): “the Brillouin spectra are indeed subjected to frequency broadening coming from the spectrometer response function, thus necessitating a spectral deconvolution to extract the intrinsic Brillouin linewidth of the sample. For this reason, it has been challenging to accurately measure the Brillouin linewidth using frequency-domain approaches²⁷⁻³²; nonetheless, when appropriate deconvolution is performed, Brillouin spectra have demonstrated strong sensitivity to the material’s longitudinal viscosity³³.”

Additionally, we added the following to the Discussion section to further address known challenges and to provide sufficient justifications for the deconvolution method we have shown in this manuscript (lines 295-302 of the revised version of the manuscript): “Accurate estimation of the Brillouin linewidth using VIPA-based spectrometers has been indeed challenging²⁷⁻²⁹. While techniques such as impulsive stimulated Brillouin scattering and time-domain approaches have been developed to achieve higher spectral resolution and more accurate Brillouin linewidth measurements, their applicability to living biological samples remains limited^{30,31,52-54}. In light of this, we sought to develop a generalizable and straightforward method to deconvolve the Brillouin spectra acquired with spontaneous Brillouin microscopes by requiring only the addition of two extra standards to the calibration protocol established in literature^{33,36}.”

2. Use of Stimulated Brillouin Microscopy (SBM):
The rationale for employing a stimulated Brillouin microscopy system is not adequately justified. While SBM offers certain advantages, it does not inherently improve spectral resolution. Furthermore, the reported use of tightly focused 100 mW laser power is concerning. No thermal analysis is provided to estimate the temperature rise at the focal spot, which is critical, especially since previous studies (e.g., <https://doi.org/10.1073/pnas.2413938121>) have shown that even significantly lower intensities can be harmful at these excitation wavelengths. Given that viscoelastic properties are temperature-sensitive, it is unclear whether the measurements obtained under these conditions reflect intrinsic cellular properties or thermally induced artifacts.

We thank the reviewer for this comment. We apologize if this point was not clear enough in our manuscript.

In stimulated Brillouin microscopy, we are able to measure the intrinsic Brillouin linewidth of materials as the spectra is free from spectral dispersive elements. In this work, we compared the linewidth acquired on the same liquids (water, methanol, N₂, and high viscosity media) using both spontaneous Brillouin microscopy and Stimulated Brillouin microscopy. By doing this, we can evaluate the relationship between the linewidths and then perform the deconvolution of the spontaneous Brillouin microscope to obtain the true linewidth. The 100 mW reported refers to the pump power of SBM, that was used only for acquiring 100 spectra of the calibration materials: here, the measurements lasted ~10 seconds, during which it is unlikely that we heated the sample.

Importantly, all cellular samples were mapped only using spontaneous Brillouin microscopy at an average power of 50 mW, using a wavelength of 660 nm, that has been proved not to induce blebbing or thermal damage to the cells (Nikolic et al., Biomed. Opt. Express 2019, doi: 10.1364/BOE.10.001567). While it is known that there are thermal concerns, we saw no changes in the Brillouin shift over time for our experiments. We never applied SBM to cells since the thermal damage, as the referee rightly pointed out, can be significant.

In order to make this point clearer, we added the following phrase in the revised version of the manuscript (lines 306-312) and cited the suggested paper about thermal effects on cells: “In such a manner, we used linewidth data obtained from SBM solely to deconvolve the spectra acquired with our BM on cells. Importantly, SBM was not applied to biological samples, as its use in this context remains limited, due to the high laser powers involved, which can cause thermal damage to cells^{32,55,56}. In contrast, spontaneous Brillouin microscopy is currently the standard technique for probing cellular and tissue biomechanics⁵⁷; it is more widely adopted, less expensive, and better suited for biological applications than SBM.”

We also specified that the acquisitions made on biological samples were acquired with the spontaneous Brillouin Microscope (lines 81-83): “In this work, we: i) propose a methodology to extract the intrinsic Brillouin linewidth of inorganic and biological samples, related to their local viscosity, from the Full Width at Half Maximum (FWHM) of their spectra acquired with our spontaneous Brillouin Microscope; ...”.

3. Imaging Protocol and Temporal Resolution: The imaging protocol lacks sufficient detail to assess the validity of the results. Living cells are inherently dynamic, and the observed spatial variations in viscoelastic properties shown in Figure 3 may reflect temporal fluctuations rather than spatial heterogeneity. The comparison with brightfield images is not convincing, as those are

acquired much faster and represent integrated axial information, whereas Brillouin images are acquired over a longer duration and are restricted to a single optical section. More thorough explanation and controls are needed to validate the stability and reproducibility of the measured properties.

We thank the reviewer for this comment and apologize if the imaging protocol was not sufficiently detailed. We agree that these living cells are inherently dynamic. Our brightfield and fluorescent images were acquired immediately before Brillouin images were acquired. Moreover, fluorescent images were acquired in confocal mode, whose focal plane has been aligned to be the same as the Brillouin focal plane, so that we were sure that the imaging plane of the fluorescence was the same as the Brillouin.

We acquired 1 XY plane of the cell at the middle Z position of the nucleus. We did not acquire multiple XY planes, making our acquisition time significantly faster. Each acquisition used only an exposure time of 0.05 s with a step size of 0.5 $\mu\text{m}/\text{pixel}$, for a total of ~2-3 minutes per acquisition. After each Brillouin acquisition, we acquired another brightfield image to ensure cells were stable, were not moving and did not show any thermal-induced damage like blebbing, which is a symptom of cell apoptosis (Nikolic et al, Biophys J 2022, DOI: 10.1016/j.bpj.2022.09.002; Li et al., PNAS 2024, DOI: 10.1073/pnas.2413938121): indeed, after the Brillouin acquisitions blebbing never happened and cells never moved. Our imaging protocol matches other Brillouin imaging protocols available in the literature with our state-of-the-art Brillouin Microscope, already described in detail elsewhere (Zhang J. & Scarcelli G., Nat. Protocols 2021, doi: 10.1038/s41596-020-00457-2).

We apologize if these details were missing in the previous version of the manuscript. In the revised version, the imaging protocol is now better described in the Methods section (lines 578-586 of the revised manuscript): “Our imaging protocol matches other Brillouin imaging protocols available in the literature with our state-of-the-art Brillouin Microscope^{33,36}. Briefly, to obtain 2D Brillouin maps of cells, YZ or XZ scans were acquired by scanning through the nucleus of the cell sample using a motorized stage (step size 0.5 -1 μm). From there, the middle of the nucleus is identified as the optimal Z-level and XY Brillouin maps of the cells were acquired. Each acquisition used an exposure time of 0.05 s with a step size of 0.5 $\mu\text{m}/\text{pixel}$, for a total of ~2-3 minutes per acquisition. We stained cells for 20’ with HOECHST and saw them live under transmission and fluorescence mode. Brightfield and fluorescence images were acquired immediately prior to Brillouin imaging; after each Brillouin acquisition, we checked the brightfield to ensure cells were stable, were not moving and did not show any thermal-induced damage like blebbing, which is a symptom of cell apoptosis^{32,47}.”

4. Data Interpretation and Analysis:

The presence of prominent "hot spots" in the Brillouin images raises questions about data consistency and interpretation. The manuscript does not address whether these features are artifacts, localized structural elements, or influenced by imaging parameters (e.g., focal plane, axial position). A discussion of their origin and potential impact on data interpretation is necessary to support the reliability of the conclusions.

We thank the reviewer for this comment. The "hot spots" present in the Brillouin maps correspond to subcellular structures seen in the brightfield images.

Some spots can be seen in the nucleus and are recognized as nucleoli, the largest structure in the nucleus, which are non-membrane bound structures made of RNA, DNA and proteins. They have been characterized by others in the Brillouin Microscopy field, showing similar spots in the Brillouin maps corresponding to the stiffest portion of the cells, as here the DNA is highly packed (Antonacci et al., Sci. Rep. 2016, doi: 10.1038/srep37217; Schlübler et al., eLife 2022, doi: 10.7554/eLife.68490).

Some other spots can be seen in the cytoplasm as little black spots and their presence is obvious in the brightfield, Brillouin shift and width maps. About these subcellular structures, we are unsure which compartment they belong to, but other Brillouin data showed the presence of the same black spots even in the brightfield and Brillouin images of other cell lines, such as HeLa cells (see Figure 3b of Qi et al., Nature Photonics 2025, doi: 10.1038/s41566-025-01697-y), NIH3T3 cells (see Figure 2k, 2l and Figure 4a of doi.org/10.1101/2025.02.26.640449), mouse fibroblast cells (Extended Data Fig 4 of doi.org/10.1038/s41592-023-02054-z). We here show some other data obtained on our MDA-MB-231 cells in which these structures can be better seen by brightfield (scale bars = 5 microns):

And also some our unpublished data of HeLa cells showing the same black spots (scale bars = 5 microns):

We can thus conclude that such spots are probably cytoplasmic structures typical of different cell lines that are not related to our imaging protocol nor are induced by the laser. Importantly, they are also visible at the brightfield images, which we always acquire before the Brillouin imaging. In any case, the presence of such structures do not impact on our analysis, as they are in the cytoplasm and we only analyzed the contribution of the nuclear Brillouin shift and width, where they are not present.

We apologize if these important comments were not available in the first version of the paper. We have now added the following to the Brillouin Microscopy imaging section of the methods section to further clarify how we obtain Brillouin shifts and FWHM for the nuclear region (lines 592-593 of the revised manuscript): “The Brillouin shift and FWHM were averaged for the nuclear region of interested identified based on the fluorescence intensity of the HOECHST signal.”

Additionally, we have added the following in the Results section to clarify the presence of these structures (lines 479-485 of the revised manuscript): “These maps also demonstrate the high spatial resolution of our Brillouin microscope, allowing us to resolve subcellular structures (visible in Brillouin shift and widths maps, as well as in brightfield images) such as the nucleoli in the nucleus^{76,77} and localized features in the cytoplasm, already detected in other cell lines⁷⁸⁻⁸⁰. These cytoplasmic features, appearing as dark spots in brightfield images captured prior to Brillouin acquisition, confirm that they are not artifacts introduced by the Brillouin imaging process.”

Overall, while the study has potential to contribute meaningfully to the field, the current manuscript requires major revisions to clarify experimental design, validate measurement techniques, and improve the rigor of data interpretation.

We thank both reviewers for their insightful suggestions and observations. We hope that the revised manuscript will be now clearer and more robust.